# Parametrization of a lakes water dynamics model MLake in the ISBA-CTRIP land surface system (SURFEX v8.1)

Thibault Guinaldo[1], Simon Munier[1], Patrick Le Moigne[1], Aaron Boone[1], Bertrand Decharme[1], Margarita Choulga[2], and Delphine J. Leroux[1]

[1]Centre National de Recherches Météorologiques, Université de Toulouse, Météo-France, CNRS, Toulouse, France
[2]Research Department, European Centre for Medium-Range Weather Forecasts (ECMWF), Reading, RG2 9AX, UK

**Correspondence:** Thibault Guinaldo (thibault.guinaldo@meteo.fr)

**Abstract.**

Lakes are of fundamental importance in the Earth system as they support essential environmental and economic services such as freshwater supply. Streamflow variability and temporal evolution are impacted by the presence of lakes in the river network, therefore any change in the lake state can induce a modification of the regional hydrological regime. Despite the importance of the impact of lakes on hydrological fluxes and the water balance, a representation of the mass budget is generally not included in climate models and global scale hydrological modeling platforms. The goal of this study is to introduce a new lake mass module, MLake (Mass-Lake model), into the river routing model CTRIP to resolve the specific mass-balance of open water bodies. Based on the inherent CTRIP parameters, the development of the non-calibrated MLake model was introduced to examine the influence of such hydrological buffer areas on the global scale river routing performances.

In the current study, an off-line evaluation was performed for four river networks using a set of state-of-the-art quality atmospheric forcings and a combination of in situ and satellite measurements for river discharge and lake level observations. The results reveal a general improvement in CTRIP simulated discharge and its variability, while also generating realistic lake level variations. MLake produces more realistic streamflows both in terms of daily and seasonal correlation. Excluding the specific case of Lake Victoria having low performances, the mean skill score of Kling-Gupta Efficiency (KGE) is 0.41 while the Normalized Information Contribution (NIC) shows a mean improvement of 0.56 (ranging from 0.15 to 0.94). Streamflow results are spatially scale-dependent, with better scores associated with larger lakes, and increased sensitivity to the width of the lake outlet. Regarding lake levels variations, results indicate a good agreement between observations and simulations with a mean correlation of 0.56 (ranging from 0.07 to 0.92) which is linked to the capability of the model to retrieve seasonal variations. Discrepancies in the results are mainly explained by the anthropization of the selected lakes which introduces high-frequency variations in both streamflows and lake levels that degraded the scores. Anthropization effects are prevalent in most of the lakes studied, but they are predominant for Lake Victoria and are the main cause for relatively low statistical scores for this river. However, results on the Angara and the Neva rivers also depend on the inherent gap of ISBA-CTRIP processes representation which relies on further development such as the partitioned energy budget between the snow and the canopy over a Boreal zone. The study is a first step towards a global coupled land system that will help to qualitatively assess the evolution of future global water resources leading to improvements in flood risk and drought forecasting.

## 1 Introduction

Only 2.5 % of the total water mass of the planet is defined as fresh water, and only a very small fraction is directly accessible for human consumption (Oki and Kanae, 2006). Lakes are of fundamental importance to ensure freshwater supply to the 800 million people which have insufficient safe drinking water, according to the World Health Organization (WHO, 2010; Marsily et al., 2018). Depending on the definition of the surface area-based lower limit, the total number of lakes on Earth ranges from 117 million to 300 million which represents 3.7 % of the non-glaciated land surfaces (Lehner and Döll, 2004; Verpoorter et al., 2014). However, lake density is not evenly distributed on the surface of the globe. Regions like Scandinavia and Northern Canada contain the majority of these water bodies (Downing et al., 2006).

Where present, lakes play a triple role in the Earth system affecting the energy and the water budgets of General Circulation Model (GCM) and inducing a modification of the local climate and hydrology (Bonan, 1995; Mishra et al., 2010; Krinner et al., 2012).

First, they influence the atmospheric boundary layer as opposed to riparian land in terms of surface energy storage. In addition, lakes influence the freshwater flux variability which in the end interact with the local (Sauvage et al., 2018) and global ocean circulation (Rahmstorf, 1995). Moreover, the inclusion of the representation of lake fluxes into numerical weather prediction models can lead to the reduction of forecast errors (Balsamo et al., 2012).

Second, as sentinels of climate change, lakes must not only be seen as water reservoirs but also as a major ecological levers. They reduce the adverse biodiversity footprint caused by climate change by acting as carbon sinks (Williamson et al., 2009; Jenny et al., 2020). Multiple studies have demonstrated the climate influence on lake surface temperatures (Wagner et al., 2012; Palmer et al., 2014; Sharma et al., 2015; O'Reilly et al., 2015). This is important since surface temperature impacts the lake ecosystem and drives the inherent lake heat budget thus the lake mixing regimes (Woolway and Merchant, 2019). The large majority of lakes are located in high latitudes which is where air temperatures have risen more than the global average over the last century (Hartmann et al., 2013). This change retroactively affects the regional climate characterized by a warming effect in autumn/winter and a cooling effect in spring (Martynov et al., 2012; Samuelsson et al., 2010; Le Moigne et al., 2016). Global climate change also constitutes a great environmental threat: volumes of several lakes among which are the Great Salt Lake (USA), Lake Chad (Chad, Cameroon) and Lake Urmia (Iran), have shrunk significantly and lead to local and regional health disasters (Wurtsbaugh et al., 2017; Gross, 2017; Pham-Duc et al., 2020). Increasing surface temperature and human pressure on lakes reduces freshwater supply and its quality, disrupting in turn the biological and physical equilibrium through contaminant pollution or reduced freshwater storage (Williams, 1996; Cai et al., 2016; Eriksen et al., 2013; Codling et al., 2018; Rodell et al., 2018).

Third, lakes interact with the regional-scale water fluxes by increasing the potential over-lake evaporation and lowering the inter-annual and seasonal variability of downstream discharge (Mishra et al., 2010; Bowling and Lettenmaier, 2010; Cardille

et al., 2004). As a secondary moisture source they can influence regional-scale climate(Krinner, 2003; Dutra et al., 2010;
Samuelsson et al., 2010) and local precipitation (Pujol et al., 2011; Thiery et al., 2015; Koseki and Mooney, 2019). For ex-
ample, Bowling and Lettenmaier (2010) showed that Arctic lakes influence spring peak flow by storing up to 80 % of the
snow-melt water, and simulations over the arctic regions demonstrated a 5 % increase of annual mean evapotranspiration over
the Great Lakes region Mishra et al. (2010). These open water bodies are large reservoirs that generally have peak storage
in spring and gradually release these volumes to sustain summer low flows. Lake hydrological effects are size-dependent and
result in a damping of flood waves in terms of magnitude and temporally shift the variability (Spence, 2006). Water dynamics
inherent to lakes are driven by their water balance and consequently by their level variations. These key variables affect most
of the internal lake processes and control their interactions with other hydrological components. Historical and projected lake
levels drops or increases have been documented (Rodell et al., 2018; Wurtsbaugh et al., 2017) and have lead to the modifi-
cations of internal processes such as lake mixing regimes and regional water availability (Vörösmarty et al., 2010; Woolway
et al., 2020).

Lakes have long been considered as a discontinuity within the river network, but there is a general agreement now that
consideration of the rivers and lakes as a continuum is required (Jones, 2010). Therefore, lakes must be taken into account
in global climate change impact studies as populations depend on their inherent ecosystem services (e.g. drinking water, fish-
ing, tourism and leisure. Schallenberg et al., 2013). Multiple studies have expressed the regional (Ogutu-Ohwayo et al., 1997;
Smith et al., 2015; Zhang et al., 2016) and global (Janse et al., 2015; Goudie, 2018) threat impacting lakes, and they reveal the
direct and indirect influence of human activities on biodiversity. The global interest of lakes has led the scientific community
to make a call to warn society about the rapid degradation of large lakes worldwide (Jenny et al., 2020). Models are frequently
used as the basis for prediction, but development of land surface models (Noilhan and Planton, 1989; Krinner et al., 2005;
Balsamo et al., 2009) and river routing models intended for large scale applications (Vörösmarty et al., 1989; Hunger and Döll,
2008) have been generally focused on overland flow, groundwater representation and river routing with less attention on lateral
fluxes (Davison et al., 2016). Among these, there was a lack of consideration of lake water mass dynamics (Gronewold et al.,
2020) because of both the coarse resolution of global models and the associated increased computational costs. Global Climate
Models (GCMs) usually consider lake energy budget without giving much importance on the river-lake connectivity, even if
key regions in climate studies such as Scandinavia and Northern America are mainly dependent on this. Global Hydrological
Models (GHMs) usually represent lakes as large rivers with modified characteristics in order to retrieve the correct downstream
river discharge. To address a comprehensive outcomes resulting from long-term water cycle evolution, GHMs need to charac-
terise every key component interacting with each other (Gronewold et al., 2020).
In the recent years, many studies have focused on anthropogenic open waters (Hanasaki et al., 2006; Haddeland et al., 2006;
Gao et al., 2012) with less attention devoted to the understanding of natural lake global influence on the global water cycle. All
this advocates for a realistic representation of lake mass balance in climate studies in order to study their role in the global water
budget in addition to flood risk management, drought predictions and in helping stakeholders to implement realistic policies
in water resource management. To our knowledge, only a few models consider specific processes driving lake mass balance

(Table 1). These models have been used for improving flood forecasting (Zajac et al., 2017), assessing the impact of lakes on river streamflows (Huziy and Sushama, 2017) and understanding the impact of open water bodies in the regional water cycle (Bowling and Lettenmaier, 2010). The main outcome of these studies is the necessity to implement lakes in a hydrological model as they affect both the regional and global water transfer. Nonetheless, even the latest research efforts remain at a coarse resolution, which limits the number of lakes that can be represented. These models are often calibrated in order to retrieve local water patterns which limit the ability to implement such schemes at the global scale. Finally and to our knowledge, no mass-balance lake models are effectively integrated within the land surface system for use in climate modelling and global hydrological applications.

As one of the contributors to the Intergovernmental Panel on Climate Change (IPCC), Météo-France's Centre National de Recherche Météorologiques (CNRM) is in charge of the development of the Climate Model called CNRM-CM (CNRM-Climate Model), the sixth version of which has been released (Voldoire et al., 2019). This climate model contains an improved representation of the coupled thermal and hydrological processes of the land surface called ISBA-CTRIP (Decharme et al., 2019). This system is based on the coupling between the Interaction-Sol-Biosphère-Atmosphère (ISBA) land surface model (Noilhan and Planton, 1989) and the CNRM version of the Total Runoff Integrating Pathways (CTRIP) river routing model (Oki and Sud, 1998; Decharme and Douville, 2007). Thanks to recent developments described in detail in Decharme et al. (2019), CTRIP is now one of the only global model representing the joint effect of floodplains and groundwater on the surface water and energy budget in a climate model. However, the representation of lakes in the model is limited to the energy budget computation by the bulk-model FLake (Mironov, 2008) which does not take lake mass fluxes into consideration.

The purpose of the study is to implement lake processes in the CTRIP river routing model. This paper will examine the impact of introducing this non calibrated lake model MLake at the global scale on river discharge. It will also assess the performance of retrieving correct water storage variations by comparing observed and simulated lake level variations. To do so, MLake has been implemented in the more recent CTRIP river routing model at a resolution of 1/12°, which is the upper limit in resolution for the physical processes in the current CTRIP model (otherwise, changes to the module formulation and introduction of hydrodynamic processes would likely be necessary). Within the system, ISBA simulates runoff and drainage in response to atmospheric forcing whilst CTRIP, the river routing model, transfers water through the hydrographic network of the resolved watersheds. Note that there are challenges to evaluating such a new model, since global lake datasets remain scarce or incomplete. This is mainly explained by the extensive detailed field measurements required, such as bathymetry profiling and the associated costs (Hollister and Milstead, 2010). This study tries to overcome these limitations by using inherent CTRIP parameters like the river channel width at the lake outlet, which obviously leads to uncertainties. Sensitivity tests are done by prescribing different outlet width configurations and then studying their impact on both the river streamflow and the lake level amplitude and variability for multiple study sites.

## 2 Modelling framework

### 2.1 ISBA-CTRIP system

The ISBA-CTRIP system (https://www.umr-cnrm.fr/spip.php?article1092; accessed: 01/09/2020; Decharme et al., 2019) simulates the surface energy and water budgets for large scale climate and hydrological applications. A schematic of this coupled model is shown in Fig. 1 Spatially distributed, this model has been evaluated globally in off-line mode (i.e. decoupled from the atmosphere and forced at the upper boundary using an optimal blend of observations and numerical weather prediction output) using two sets of atmospheric forcings against in situ measurements and satellite products. The most significant results show improvements in the river discharge simulations, the snowpack representation and the land surface evapotranspiration (Decharme et al., 2019). Recently, the updated version of ISBA-CTRIP, considering improvements such as wild fires and land cover changes (Séférian et al., 2019), has also shown a better representation of global-scale carbon pools and fluxes (Delire et al., 2020).

Originally the land surface model ISBA simulated several key land surface variables such as surface runoff or soil moisture in response to atmospheric forcings based on a force-restore approach. This scheme represents land processes as a single soil-vegetation-snow continuum limiting the prediction of root layer droughts and the heterogeneity of soil properties. Currently, the diffusive version of ISBA is used for hydrological and climate modelling applications. It explicitly resolves both the one-dimensional Fourier and Darcy laws for subsurface thermal and mass fluxes, and it accounts for the hydraulic and the thermal properties of soil which is now discretized in 14 layers resulting in a total depth of 12 m. In addition, the scheme can include the effects of soil organics on the thermal and hydrological properties of the soil. The snow is simulated using a multi-layer snow model based on the work of Boone and Etchevers (2001) which recent improvements in physics and increased vertical resolution as described in Decharme et al. (2016).

ISBA is fully integrated within the surface modelling platform SURFEX (v8.1) (Masson et al., 2013; Le Moigne et al., 2020) developed at the CNRM in order to bring all the models related to the surface parametrisation into one unique software platform. SURFEX allows studies to be performed in offline mode or fully coupled to an atmospheric model extending -de facto-its applicability range from local hydrological to large scale climate studies. The distinction of such land processes in SURFEX comes from the global land cover database ECOCLIMAP-II which dynamically renders the type of vegetation and its cover at the chosen spatial resolution of the model for a given application (Masson et al., 2013; Faroux et al., 2013).

ECOCLIMAP-II is a 1 km resolution land-use and land-cover database based on satellite products designed for operational and research numerical weather prediction, climate modelling, hydrological forecasting, and in land surface numerical studies within the SURFEX surface modelling platform (Le Moigne et al., 2009). ECOCLIMAP-II details whether a pixel contains one of the four different type of covers: lake, town, land or ocean, and it distinguishes hundreds of plant functional types representing a large variety of ecosystems (Faroux et al., 2013). SURFEX further aggregates the initial covers into upwards of

20 patches which correspond to different land covers or plant functional types. The orography is extracted and upscaled from the 90 m resolution Shuttle Radar Topography Mission to a 1 km-resolution (Werner, 2001). The ECOCLIMAP-II lake cover scheme provides binary information on the presence or not of a lake in the pixel. No other information is provided thus lake cover information is completed with the Global Lake DataBase (GLDB, Kourzeneva et al., 2012; Choulga et al., 2014) which has gridded in situ and estimated lake mean depth at 1 km resolution globally. This global database has been developed to gather lake information and retrieve mean depth information for numerical weather prediction. It already serves as input for correcting land cover used by SURFEX for approximately 15,000 lakes on a 1-km resolution grid. However, a dataset threshold is introduced on lake detection and set at a surface area of 1 km$^2$ that limits the number of lakes considered in our calculations. In this research, we used continuous mean depth field recently developed at ECMWF (Choulga et al., 2019) to ease aggregation technique from 1-km to 1/12°.

Streamflow routing is simulated using CTRIP (Fig.1) which integrates a dynamic computation of river flows based on a kinematic wave approximation which is solved using Manning's roughness equation as a friction energy dissipation term which is dependent on the characteristics of the river section. CTRIP is fully coupled to SURFEX and considers the interaction between the rivers, the atmosphere and the soil through the input by CTRIP which then computes the river discharge, water table evolution and surface flooded fraction. Moreover, it explicitly accounts for groundwater processes with the integration of a two-dimensional diffusive aquifer scheme connected to rivers and a parameterization of the capillary fluxes within the soil (Vergnes et al., 2014). Descriptions of the parameterization of flooding processes can be found in Decharme et al. (2019). The coupling of ISBA and CTRIP is made through the OASIS3-MCT coupler (Voldoire et al., 2017) where ISBA provides surface runoff and drainage estimates which are then transformed by CTRIP in river discharge, water table height or floodplain fraction. In addition to the fully coupled configuration, CTRIP can be used in an offline configuration forced by the runoff/drainage coming from ISBA (or other land surface model) simulations and without feedbacks between the water bodies and the soil processes. Further details on the physical processes are presented in Decharme et al. (2019).

In this study, we refer to CTRIP as a global scale model meaning that it is a 1/12° degree resolution model applied to areas ranging from large basins to a domain covering the entire globe.

Each CTRIP pixel represents an unique rectangular river section with its own characteristics. As shown in Fig. 2, instead of working directly with grid cell, each river section is integrated as a node in the network and all nodes are labelled sequentially. Their number defines the position of the river section in the network for each hydrographic basin. The scheme increasingly iterates on this number and ensures all the upstream masses have been updated before the numerical computation on a designated node of the network starts. This numerical solution framework assures the computation of river discharge is performed starting from the upstream cells and then progressing to the downstream cells of the watershed. In every basin, the head-water cells have the lowest sequence order: one, which is incremented for each downstream cell. The general rules of attribution consider that a node can receive water from multiple affluents but can not have multiple downstream sections. Considering the case of an affluent with multiple upstream nodes and in order to avoid conflicts at the confluence, the downstream sequence

order $SN_{downstream}$ attribution follows the rule:

$$SN_{downstream} = max(SN_{i,upstream}) + 1 \qquad\qquad i \in [1, N] \qquad\qquad (1)$$

where $SN_{i,upstream}$ represents the sequence number of the upstream river, $i$, and $SN_{downstream}$ is the sequence number of the downstream river.

The main motivation for the integration of new processes in CTRIP is to both simulate river discharge and to enable the quantification of the impact of climate change on drought and flood risk over the entire globe. It is also a valuable tool which gives estimates of global water resources in the context of global depletion. Regarding the global water budget, the ISBA-CTRIP model improves the simulations of both peak discharges and baseflow, in addition to global terrestrial water
storage variations. However, Decharme et al. (2019) addressed the need to increase the resolution in order to avoid a subgrid parameterization and in order to consider the water dynamics more precisely. Originally used at a resolution of 1°, then down-scaled at 0.5°, on-going improvements permit the model to run at its current resolution: 1/12° (approximately 6-8 km at mid-latitudes). This resolution guarantees a better discretization of surface and subsurface processes without the need to implement additional river hydrodynamic processes. The river network at 1/12° has been derived by applying the Dominant
River Tracing algorithm (DRT; Wu et al., 2012) on the high-resolution river network (3 arcsec) of MERIT-HYDRO (Yamazaki et al., 2019). CTRIP parameters describing river properties as well as floodplain and aquifer characteristics have been derived following the same methodology as for the 0.5° version of CTRIP (for details see Decharme et al. (2019)).

### 2.2   Flake: A lake energy balance model

Lake evaporation is simulated using the FLake model (Mironov, 2008). When considered together with the precipitation, an
estimation of water mass exchange by the lake with the atmosphere can be made. FLake is a bulk-model capable of simulating the lake energy budget within the lake and at the lake-atmosphere interface (Mironov et al., 2010). FLake is designed mainly for use in numerical weather prediction and climate studies where it helps determining the vertical lake temperature structure, the mixing conditions and the retroactions with the local/regional climate (Balsamo et al., 2012; Le Moigne et al., 2016; Salgado and Le Moigne, 2010). FLake is based on a numerical solution of a two-layer parametric evolution of the temperature
profile and the integral budgets of heat and kinetic energy. The mixed layer is characterized by a uniform temperature and an entrainment equation which estimates the layer depth. Below this first layer, the vertical temperature profile is parameterized in order to represent the thermocline shape based on a self-similarity concept (Kitaigorodsky and Miropolsky, 1970). This model uses external parameters of which the most important ones are the lake mean depth and the extinction coefficient (set to 0.5 m$^{-1}$ following Le Moigne et al. (2016)). The numerical solution is based on the evolution of four lake prognostic variables:
the surface temperature, the lake bottom temperature, the thickness of the mixed-layer, the shape factor and one parameter: the mean lake depth. An extensive description of the model can be found in Mironov (2008).

## 2.3 MLake: A global scale mass balance lake model

### 2.3.1 Generation of a global lake mask

Before implementing the numerical representation of lake dynamics into the CTRIP model, lakes need to be introduced in the river network at 1/12°. However the ECOCLIMAP-II provides binary information of the lake detection at 1/120° meaning the information needs to be upscaled to the CTRIP resolution. The method is based on a recursive aggregation of neighbouring lake pixels which depends on the GLDB mean depth. In other words, for every pixel at 1/120°, the algorithm scans the surrounding pixels and aggregates those that are connected and have the same mean depth. Each aggregated lake is then identified with a unique number used further when attributing inherent parameters and variables.

This method is developed for large lake identification but struggles in the regions with a high density of small lakes, e.g. Finland. For example, estimated lake mean depth in all Boreal zone is based on geological method taking into account a tectonic-plate map and geological maps (Choulga et al., 2014). The geological method assumes that lakes of the same origin and region should have the same morphological parameters, e.g. mean depth. In our study small lakes tend to be aggregated as a unique larger lake that might not represent the local morphology. These anomalies can modify the local hydrology, however considering the scale of the current study, these effects are limited or even can be filtered by averaging.

### 2.3.2 Integration of lakes in the river routing model (RRM)

At the model resolution of CTRIP, a unique river stretch is attributed to each grid cell. Replacing a river pixel by a lake follows the same logic as water transfer, which is dependent on the riparian topography and its location within the watershed. However, integrating a lake which can cover more than one grid cell in the CTRIP river networks is not straightforward. Huziy and Sushama (2017) proposed a distinction between local lakes, covering at least 60 % of a grid cell, and global lakes, which can cover several grid cells. This distinction brings some dynamic limitations as a local lake can only be an extension of the river section that contributes to the downstream flow without being fed by the river itself. On the contrary, a large lake is part of the river network and divides the river in an upstream section that contributes to the total lake inflow, and a downstream section connected to the lake, receiving its mass from the lake outlet. However, it is important to keep a unique method that can adapt to all lakes regardless of their size.

Some issues related to the integration of lakes in the river network emerge when considering that lakes add a spatial dimension to the network linked to the fraction of pixel covered. First, the model must estimate a correct partitioning of the runoff between rivers and lakes when both components are located on a pixel. At the 1/12°, a lake can cover a small fraction of the pixel while being actually part of another watershed. This is the case for the Lake Bourget (France, Fig. 4) where a river that flows on another watershed contains most of the runoff of the pixel whilst the lake only captured a small amount of water that is part of the lake watershed. The other issues concern the location of the lake in the river network and which river stretch is actually a part of lake. In some regions, the river stretch can be large and thus the streamflow time response remains slow which can

be close to the response time of a lake. Consequently, finding a compromise between the lake spatial extension at different resolutions and the actual lake water dynamic is important. The approach used herein to resolve this issue is to replace a river section by a lake pixel (corresponding to a unique node in the network) when a lake covers at least 50 % of a given grid cell (Fig. 2). Wherever a lake spreads over several grid cells, two distinct lake masks are necessary. This is important, on the one hand, to ensure that the water flux remains realistic and on the other hand, the introduction of lake mass dynamics should not significantly change the local hydrology.

First a lake mask, called the "network mask", is needed to locate the lake within the river network and to link the considered lake to the correct river. The procedure of this integration is based on the steps presented on the Fig 3. In CTRIP, an identification number is assigned to every river that allows a distinction between rivers of the same watershed. This identification number comes from the upscaling of the 90 m resolution MERIT HYDRO (Yamazaki et al., 2019). The upscaling of the river network from 90 m to 1/120° resolution preserve the continuity of this ID number. Identification of lake pixels follows the same rules. To do so, a function recursively determines every lake pixel at a 1/120° resolution that covers a river stretch of the MERIT HYDRO river network with the same identification number as the river that flows at the outlet (river stretch identified in yellow on the Fig 3b). Thereby, every lake pixels are linked to the correct river ID number and this link is preserved while upscaling at the 1/12° resolution (Fig 3d).The network mask ensures that all of the lake pixels with the same ID number are coupled within a unique mass-balance process. However, as shown on the Fig 3d, few conflicts may appear while applying this method. On this particular example, the northern pixel is not part of the lake's watershed and flows out within another basin which induce conservation issue. A second function recursively determines every lake pixel at a 1/12° resolution that covers a river stretch of the CTRIP river network (river stretch identified in pink on the Fig 3d). This last step ensures the lake network only considers lake pixels that are effectively in the river basin. The lake network mask for lake Bourget is shown on the Fig 4a. At the end of each time step, diagnostic variables are distributed on this mask. This method ensures all freshwater lake pixels are effectively linked to the correct river within the entire network and that water mass flowing in a different watershed is not entering the lake.

Thus, a second lake mask is needed, the lake runoff mask. The runoff mask creation is based on the lake information coming from ECOCLIMAP at 1/120° resolution as presented on the Fig 4b. In fact, this runoff mask corresponds to every CTRIP pixel at the 1/12° resolution that contains at least one ECOCLIMAP lake pixel (at the 1/120° resolution). In other words, this is a mask of the lake fraction at 1/12° without any distinction of the watershed nor the lake fraction. It provides information on the spatial extension of the lake within the river network, and it is used for computing the water mass intercepted by the lake from the LSM (as runoff and drainage).

### 2.3.3   Lake model

The MLake mass balance equation is based on the difference between the mass fluxes entering and leaving the lake. At each time step, the lake module calculates the prognostic net water storage $V_{lake}$ $(kg)$ over the lake surface area based on the

following equation:

$$\frac{dV_{lake}}{dt} = P_{ol} - E_{ol} + R + D + Q_{in} - Q_{out} - Q_{gw} \tag{2}$$

where $t$ is the time (s), $P_{ol}$ is the over-lake precipitation term (kg.s$^{-1}$), $E_{ol}$ is the over-lake evaporation term (kg.s$^{-1}$), $R$ and $D$ are terms to account for respectively runoff and drainage estimated by ISBA (kg.s$^{-1}$) over the runoff mask, $Q_{in}$ is the inflow entering the lake from the tributaries (kg.s$^{-1}$), $Q_{out}$ is the lake outflow (kg.s$^{-1}$) and $Q_{gw}$ represents the contribution of the lake-groundwater fluxes (kg.s$^{-1}$).

The mass balance equation is numerically resolved in two steps: first an estimate of the incoming flows is computed and used to define an intermediate lake volume $V_{lake}^*$. Next, the outgoing water flow estimation based on this intermediate state in order to return to a new lake equilibrium state. Incoming flows consist in the contribution of both the riparian banks and the direct river inflows. The riparian bank runoff and drainage volumes are collected by the lake and computed over the runoff mask as shown in Fig. 3 following the rules:

$$\begin{cases} R = \sum_p r_S(p) \\ D = \sum_p d_S(p) \end{cases} \tag{3}$$

where $r_S$ and $d_S$ represent the runoff and drainage fluxes, respectively, over the pixel $p$ on the runoff mask $\omega$. The specific inflows flowing into the lake are composed of all the upstream tributaries (with a lower sequence number) connected the network mask following the equation:

$$Q_{in} = \sum_k^l q_{in}(k) \tag{4}$$

where $q_{in}$ is the river discharge of the tributary number $k$ and $l$ is the total number of tributaries for the considered lake. Even if it is not applicable for long-term hydrological analysis, and due to a lack of knowledge on the large-scale process, the groundwater flux is often the missing term indirectly retrieved from the residuals of the mass balance computation. The lateral and vertical groundwater fluxes are very sensitive to the spatial resolution (Reinecke et al., 2020). Groundwater-lake interactions are generally better-understood locally (Bouchez et al., 2015), but the representation of such interactions at a larger scale can be difficult owing to a lack of understanding of the processes involved. As a consequence, only groundwater-river processes already present in the model are activated, meaning there is no interaction between groundwater and lakes which will be integrated in a further version of MLake.

As mentioned previously, the outflows are calculated considering an intermediate lake state in order to retrieve the final lake volume. This intermediate state for the time step (s) is defined as an intermediate volume $V_{lake}^*$ (kg):

$$V_{lake}^*(t) = V(t - \Delta t) + [P_{ol}(t) - E_{ol}(t) + R_S(t) + Q_{sub}(t) + Q_{in}(t)] \Delta t \tag{5}$$

where $\Delta t$ is the time step (s) and $V(t - \Delta t)$ is the lake volume at the previous time step $t - \Delta t$ (kg.s$^{-1}$). Eq. 6 provides an estimation of the intermediate lake hydraulic head $h^*_{lake}$ (m):

$$h^*_{lake}(t) = \frac{V^*_{lake}(t)}{A_{ECO}} \tag{6}$$

with $A_{ECO}$ the lake area in the ECOCLIMAP-II database (m$^2$).

The outflow is, by definition, linked to the lake water storage assuming a rating curve relation based on an empirical weir relationship that links the discharge to the water head over the crest (Eq. 7). The outflow starts as soon as the lake height exceeds the weir height. The discharge is then function of a hydraulic head which represents the height of water above the weir. This approach mimics the lake outlet dynamic as a waterproof basin which flows out through a counter-slope. The need to model outflow at the global scale restricts the complexity of the parametrization as it needs to take into account all lake types. At the current resolution of the model (i.e. 1/12°), the outlet is assumed to be small enough to be considered as a straight section connected to the downstream river without any friction and to have the same shape as the downstream rectangular river section. This approach is represented in the Figure. 6

The outflow is calculated as:

$$Q_{out} = \begin{cases} 0 & \text{if } h^*_{lake} \leq h_{weir} \\ C_d \sqrt{2g} W_{weir} \rho_\omega (h^*_{lake} - h_{weir})^{\frac{3}{2}} & \text{if } h^*_{lake} > h_{weir} \end{cases} \tag{7}$$

where $C_d$ a dimensionless coefficient related to the drag of the weir which is prescribed to be $0.485$ (Lencastre, 1963), $W_{weir}$ the width of the outlet equal to the width of the river in the downstream pixel (m), $h_{weir}$ the height of the weir (m) and $\rho_\omega$ is the volumetric mass of the water (kg.m$^{-3}$).

The river width was first determined over France by comparing the mean annual discharge measurements from the Banque Hydro database (www.hydro.eaufrance.fr) and the river width of the "Systeme Relationnel d'Audit de l'Hydromorphologie des Cours d'Eau" (SYRAH) which leads to the empirical equation (Vergnes et al., 2014):

$$\omega_{river} = \alpha Q^\beta_{mean} \tag{8}$$

where $\alpha$ and $\beta$ are dimensionless parameters, respectively, equal to 5.41 and 0.59 (Vergnes et al., 2014). $Q_{mean}$ is the mean annual discharge of the river calculated over the climate period (1981-2010). This empirical exponential function has been extended to the global scale by Decharme et al. (2019) based on the comparison of two datasets: the Global Width Database for Large Rivers (GWD-LR: http://hydro.iis.u-tokyo.ac.jp/ yamadai/GWD-LR/) and the Global Lakes and Wetlands Database (GLWD, http://wp.geog.mcgill.ca/hydrolab/glwd/, (Lehner and Döll, 2004).

The initial lake level is equal to the weir height, which results in an initial lake outflow equal to zero. Eq. 7 incorporates the dependence of the depth on the hydraulic head over the weir. The final lake volume for the time step ($t$) is derived from the equation:

$$V_{lake}(t) = V^*_{lake}(t) - Q_{out}(t)\Delta t \tag{9}$$

Eq. (2) calculates a change in lake water storage from which the diagnostic variables, such as surface area and lake level, are estimated. Numerous hydrological models assume the lake storage to be a linear function of the surface area and depth. This solution does not take into account the specific lake bathymetry, and it simulates a realistic hypsographic relation, thus the lake surface area is supposed constant. However, knowing how the lake surface area varies with respect to depth is important for improving over-lake evaporation estimations. With regards to the relative scarcity of global-scale datasets on lake bathymetry, implementing appropriate lake hypsometric curves would require extensive developments that will be carried out in further studies. For simplicity, in the current study, hypsometric curves are assumed to be linear.

## 3   Study sites

Four watersheds have been selected in order to assess the impact of lakes on the regional scale hydrology. A map showing the location of the basins is presented in Fig. 7. They have been chosen based on several criteria: the size, their localisation in the drainage basin, and the climate characteristics (in order to assess the sensitivity of the model to different forcings conditions). These characteristics are summarized in Table 3. The first watershed is the Rhône basin with its outlet located at Beaucaire (France). Flowing from the Furka glacier in Switzerland to the Mediterranean Sea (Rhône Delta), the basin represents 17 % of the French metropolitan area. The Rhône is a socio-economic lever in terms of both quantitative (freshwater resource, industrial needs, sailing ...) and qualitative resource management (ecological state, tourism ...). In its upstream part, the streamflows are dependent on the glacier water supply whereas in its downstream part, the Mediterranean climate directly impacts the discharge and water level associated with flash flood risks. Therefore, these diverse forcings induce a bi-modal hydrological regime. Within this watershed, five lakes are identified at the spatial resolution which must be resolved within the current study, among them Lake Geneva which is one of the largest European freshwater reservoirs with an average volume of 89 km$^3$. With a relatively small drainage area compared to other lakes, Lake Geneva creates a link between the mountainous upstream and the fluvial downstream regimes. Located on the upstream part of the Rhône network, it also controls the streamflows and limits the floods during spring. Due to the importance of karstic structures on the downstream Rhône river and especially on the baseflow, this basin is the only study site where the groundwater scheme has been activated.

The second watershed is the Angara river basin in Irkoutsk (Russia). The water mass flowing from Lake Baikal controls the streamflows of the Angara watershed which flows to its confluence with the Yenissei river at Strelka. This watershed was selected in order to study the specific hydrological conditions of Lake Baikal which waters freeze in winter and its prevalence on the regional hydrological system. Known both for its unique endemic ecosystem and its morphometric characteristics,

Lake Baikal is the deepest lake in the world (maximum depth of 1,632 m) and the second largest lake in terms of volume (approximately 23,600 km$^3$). One of the lake's characteristics is its surface freezing period (approximately five months) which contributes to its specific hydrological regime.

The third watershed is the upstream part of White Nile river in Jinja (Uganda). Characterized by a dry continental climate, the White Nile originates from the outflow of Lake Victoria, which is the world's second largest lake in terms of surface area

(69,485 km$^2$). In contrast to lakes such as Lake Baikal, Lake Victoria has a relatively small drainage area (167,000 km$^2$) and its water balance is driven mainly by the precipitation and evaporation (Vanderkelen et al., 2018). Surrounded by the Great Rift Valley, it is a major socio-economic resource which directly supplies 30 million people and indirectly, over 300 million people living nearby the Nile. Since 1951, the outflow has been regulated by the Nalubaale dam, and a second dam was built in the 1990s by the World Bank. However, the regulation is controlled by an 'Agreed Curve' which intends to mimic natural outflow

and links the water releases to the lake levels.

The last watershed is the Neva river basin close to Saint-Petersburg (Russia). This relatively small river (74 km) is the main outlet of Lake Ladoga, the largest European lake. Neva is influenced by the Svir river, at the outlet of lake Onega, which is the second largest European lake. The surface area of these lakes are 17,800 km$^2$ and 9,800 km$^2$ (Filatov et al., 2019), respectively. The Ladoga hydrographic basin is complex and represents dozens of lakes that buffer the streamflows within the

400 basin. In addition, these lakes are located in the boreal zone, which are regions where the positive air temperature anomalies are the largest. Ladoga remains partly ice-free until early winter (the freezing season extends from November until the end of May), and therefore it has a significant impact on the regional meteorological conditions such as the enhancement of severe convective snowfall episodes (Eerola et al., 2014). In response, the water temperatures of the lakes, specifically those from the Onega, are sensitive to atmospheric changes because of their relatively low heat capacity (Filatov et al., 2016). The Ladoga

drainage area is approximately 97,800 km$^2$ and the Onega's is 51,540 km$^2$. These lakes are particularly affected by changes in river runoff and studies show a decline in the lake levels owing mainly to a regulation of its flows (Hanasaki et al., 2006) and complex interactions with permafrost thawing due to climate change (Karlsson et al., 2015).

## 4    Materials and Data

### 4.1    Lake observations and discharge data

Model lake level validation is based on the comparison of simulations with multi-mission satellite measurements. The elevation data come from the Hydroweb platform (available at:http://www.hydroweb.theia-land.fr, Crétaux et al., 2011). This platform provides, with centimetric accuracy, user-friendly altitude measurements for approximately 1000 sites for major rivers and approximately 230 lakes since 1993. In addition, Hydroweb provides lake surface extent and volume variations in several areas worldwide.

Some lakes are not monitored from space and in situ measurements remain the most accurate source of information. In the case of Lake Geneva, data from three measurement sites were provided by the EAWAG/EPFL institute and the Environmental Swiss Office. These observations cover the time period 1973 to 2013 and are used to monitor the level variations of Lake Geneva on

three different shores.

Regarding discharge data, a comparison was made with a dataset comprised of data from the Global Runoff Data Center (GRDC; www.bafg.de/GRDC/EN/Home/homepage_node.html), ARCTICNET and the French Banque Hydro databases (www.eaufrance.fr). From these datasets, chosen stations must have a minimum of 3 years of continuous measurements during the simulation period for a drainage area covering at least 1000 $km^2$. In the validation stage, the most downstream measurement station is chosen for comparison. But if only one station is available for the entire study site, the closest available CTRIP pixel

on the river is considered. These datasets remain incomplete and some basins lack data, such as the White Nile watershed. Lake Victoria watershed does not have any accessible discharge measurement sites. In this particular case, outflow measurements from Vanderkelen et al. (2018), who studied Lake Victoria water balance, from the Jinja Station were provided over the period 1950-2006 (Inne Vanderkelen, pers. comm.).

### 4.2 Atmospheric forcings

It is known that biases can emerge in simulated surface/sub-surface variables in response to specific atmospheric conditions, therefore different forcing datasets were used in the study. More specifically, an extensively validated high-resolution atmospheric forcing over France was preferred to coarser global forcing that may influence hydrological responses in a negative way, especially considering the large topographic variability over France. This limits the comparison between watersheds situated in France and other basins, but it gives more credit to the results between similar watersheds.

### 4.2.1 Reanalysis over France

Safran-ISBA-MODCOU (SIM, Habets et al., 2008; Le Moigne et al., 2020) is a hydrometeorological model system that results from the collaboration between the CNRM and Mines ParisTech (Etchevers et al., 2001). The system is composed of the meteorological analysis system SAFRAN (Durand et al., 1993; Quintana-Segui et al., 2008), the land surface model ISBA and the hydrogeological model MODCOU (Ledoux et al., 1989).

SAFRAN provides an analysis, based on optimal interpolation, of near-surface variables such as daily precipitation, 2-meter relative humidity, 2-meter air temperature, 10-meter wind speed, cloudiness and models visible/infrared radiative fluxes. The ISBA model is driven offline by SAFRAN analysis, and it computes the energy and water budgets in order to generate surface runoff, total evapotranspiration, soil moisture and drainage at an 8 km horizontal resolution. MODCOU uses surface runoff and drainage as inputs for river routing and aquifer water head simulations, respectively, over all of France. SIM also needs

physiographic parameters that describe the land cover, soil texture and orography of the studied zone. These parameters are provided by the ECOCLIMAP-II database.

This physically based system has several applications in operational, research and climate services: it is used in flood risk forecasting, water resource management and climate projections (Soubeyroux et al., 2008). Further details about the model can be found in Le Moigne et al. (2020). For the current study, SAFRAN and ISBA have been used to retrieve surface runoff and

soil drainage estimations for each CTRIP pixel of the Rhône watershed over the period 1958-2016.

#### 4.2.2 Global scale atmospheric variables

Uncertainties associated with the forcing variables are commonly quantified by using set of multiple atmospheric forcings. For example, (Decharme et al., 2019) used two state-of-the-art forcings for the evaluation of the ISBA-CTRIP model at the global scale. First, the Princeton Global Forcing (PGF; https://rda.ucar.edu/datasets/ds314.0/; Sheffield et al., 2006) was used over the period 1978-2014. This hourly dataset is derived from the NCEP-NCAR reanalysis for atmospheric variables (https://psl.noaa.gov/data/gridded/data.ncep.reanalysis.html) combined with the monthly gauge-based observations from the Global Precipitation Climatology Center (GPCC). Second, the Tier-2 Water Resources Re-analysis (WRR2) reanalysis from the Earth2Observe (E2O) project was used. The E2O reanalysis come from the ERA-Interim reanalysis products (https://www.ecmwf.int/en/forecasts/datasets/reanalysis-datasets/era-interim) over the period 1979-2014. Precipitation is adjusted using the monthly observations from the Multi-Source Weighted-Ensemble Precipitation (MSWEP, Beck et al., 2017) dataset. Decharme et al. (2019) showed the better performance of the model using E2O forcings compared to PGF forcings, in particular in terms of river discharge scores mainly due to higher precipitation rates. The runoff estimation for the Angara, White Nile and Neva watershed used in the current study therefore come from the multi-layer diffusive ISBA forced by the ERA-Interim E2O forcings.

## 5 Results

This study follows a two-step evaluation by assessing, first, the influence of lakes on the CTRIP streamflows simulation and second, the influence of the lake module on the performance of the model to retrieve streamflows and lake levels compared to the observations. In the following part of the results, particular attention has been paid to the model sensitivity to the lake outlet width, which is the only adjustable parameter.

### 5.1 Impact of lakes on the ISBA-CTRIP simulations

A benchmark study to evaluate the influence of the new lake module on CTRIP-simulated streamflows was first performed consisting in four simulations which are summarized in Table 4. Due to the model sensitivity to the values of the weir height, a few years of model spin-up are required to reach a steady state (the length of the spin-up depends upon the lake size). This adjustment period is not included in the evaluation. The evaluation period ranges from 01-01-1983 to 31-12-2013. The comparison of the model simulations over the period 2000-2003 is shown in Fig. 8). A general reduction of river discharge variability is observed which is associated with a delay in reaching peak discharges. With the exception of Lake Victoria, lakes have relatively little impact on the time-averaged river discharge, however they significantly reduce the river discharge variability and timing compared to reference simulation $ctrip\_nolake$. The average variability reduction over the four study sites is about 46 % (see Table 5 for a statistical summary of the benchmark runs) of the average discharge for the evaluation period 1983-2013. There is a clear scale-dependence as larger lakes have stronger impacts on streamflows. For example, Lake Geneva reduces the Rhône river discharge variability on average by 22 %, while the Angara river mean discharge decreases by 63 % due to the

influence of Lake Baikal. This is explained by the contribution of the lake to the river: the Angara river is directly influenced by Lake Baikal outflows and has no other tributaries before the gauge station in Irkoutsk. In contrast, approximately half of the Rhône discharge contributions at Beaucaire come from the part of the Rhône river flowing out of Lake Geneva, and while the remaining half comes from tributaries (Saone, Isere, Durance) which are not influenced by Lake Geneva. The implementation of lakes tends to smooth the hydrograph, reduce the volume of water transferred downstream during flood events, and increase low flows while conserving approximately the time-averaged discharge (see Table 6). Among the four study sites, the Angara and the White Nile are the most impacted rivers with a decrease of the variability that reaches 55 % and 63 %, respectively.

These results show the sensitivity of the streamflow simulations in relation to the outlet width. As expected, the outlet modulates the water volume that flows into the river by diminishing the response time of the lake to the forcing (Fig. 8). More specifically for Lake Baikal, the variability is increased by 105 % in a configuration where the weir width is increased by a factor of five compared to $ctrip\_mlake\_w1$. On the other hand, the weir width has little impact on the streamflow simulations of the Rhône river (the average standard deviation changes 3 %). However, increasing the outlet width improved streamflow dynamics and produced the discharge time series with the strongest decrease in the low-flow period and with quicker responses to the forcing in flood period. This behaviour can also induce a phase shift between outflows and inflows resulting in a period of no flows as seen for Lake Victoria in Fig. 8). Results for Lake Ladoga reveal a counter-intuitive pattern since the introduction of lakes produces an early peak discharge (both in terms of high and low flows) instead of delaying them. Flood waves take some time to propagate through the river while no time delay has been considered for lakes. Combined with a wide weir (with high flow capacity), this tends to make flood waves propagate faster.

## 5.2 Comparison of simulations to observations

In this section, the influence of the lake on the CTRIP model has been assessed by comparing both lake water levels and river discharges to measurements. In this context, the three simulations for each study site $ctrip\_mlake\_w05$, $ctrip\_mlake\_w1$ and $ctrip\_mlake\_w5$ were used with the same characteristics.

### 5.2.1 Lake water levels

In a basin where several lakes are present, the main lake, defined as the largest lake in terms of both drainage and surface area, is considered. Lake level outputs from the model are constant over all the network lake mask. Due to the initialisation method (the height of the lake crest is equal to the mean depth of the lake), the diagnostic only indicates level variations over an equilibrium level assumed to be reached after a transitory time period. Variations have been assessed by centering these levels on the time-averaged levels of the lake over the period 1983-2013. Lake level variations are shown in Fig. 9.

All of the simulations for Lake Geneva, except for $ctrip\_mlake\_w05$, show an inability to capture the range of level variations. This is due to peak levels that remain higher than observed levels. Even though the range is not correct, the model captures the seasonal variability with high lake levels associated with snow melting in spring, decreasing levels through summer/autumn

and low flows in winter. Moreover, the minimum flow values are better represented in terms of magnitude compared to the peak discharges. Regarding lakes high levels, even if the timing is acceptable, simulations show a systematic over-estimation which can reach 1m for $ctrip\_mlake\_w05$. In terms of scores as shown in the Table 7, the correlation remains low ($\bar{r} = 0.28$) which gives the impression of a weak model performance on retrieving lake levels. Standard deviations show a relative over-estimation of the level $\alpha = 2.3$ ($\overline{\sigma}_s = 0.51$ m, $\sigma_o = 0.22$ m). Along the same lines, the errors are about $0.51$ m (interval = [0.27-0.75]) confirming the systematic over-estimation. The Taylor diagram (Fig. 10) gives information on the better performance of the $ctrip\_mlake\_w5$ configuration which shows skill in retrieving both lake level variability and magnitude with a standard deviation ratio of $\alpha$=0.9, while both $ctrip\_mlake\_w1$ and $ctrip\_mlake\_w05$ do not properly simulate the observed lake dynamic ($\alpha = 2.4$ and $\alpha = 3.5$, respectively). The underlying reason is that the weir width is impacting the lake level dynamics with a level variability inversely proportional to the lake outlet width. This is physically correct as a larger outlet results in an attenuation of the time needed to transfer the mass from the entry of the lake to the outlet where a smaller outlet increases the retention capacity and the response time of the lake to the forcing. Likewise, the drainage area of Lake Geneva is relatively small thus the concentration time is small which results in a rapid response of the water dynamic to the regional forcings. Last but not least, anthropization can have a significant impact on streamflow within the Lake Geneva basin, in addition to the lake itself since it is regulated by the Seujet dam in Geneva.

In contrast, the model results are much better for Lake Baikal and Lake Ladoga in terms of the seasonal variability and the timing of peak and low flows. For Lake Baikal, results are particularly good before 2002, year when a slight shift begins. The correlation is improved ($\bar{r} = 0.76$) and standard deviations show the same degree of dispersion between observed and simulated data ($\overline{\sigma}_s = 0.26$ m, $\sigma_o = 0.28$ m). The relative variability is relatively high $\alpha = 0.93$, which shows the ability of the model to capture the seasonality and range of Lake Baikal level variations. The Taylor diagram shows the weaker performance of the $ctrip\_mlake\_w5$ configuration in terms of retrieving Lake Baikal level variations compared to the other simulations ($ctrip\_mlake\_w05$; $ctrip\_mlake\_w1$). Similar results can be seen for Lake Ladoga levels. However, simulations have a systematic temporal shift which induces both early low and high water levels in the lake. Thus, this temporal shift reduces the real performance of model by lowering drastically the correlation. Even if the amplitudes are generally well captured, a slight under-estimation of high water in 1994-1995 can be noticed as well as an under-estimation of the 2003 low levels. Even though results or the standard deviations are reasonably good ($\overline{\sigma}_s = 0.22$ m, $\sigma_o = 0.26$ m) with a relative variability $\alpha = 0.85$, the time shift degrades the correlation which is solely of 0.36 which confirms the visual agreement (figure 9). Regarding the inter-comparison of the different lake configurations, $ctrip\_mlake\_w05$ is the model that performs the best for this particular lake.

Results on Lake Victoria slightly different compared to the other three lakes, with an improved ability of the model to capture lake level variations until the period 2004-2005. After these years, a gap in the observations appears with a sharp decrease of the observed lake levels which reaches a new steady state at the end of 2006. After 2006, the variability of both simulated and observed levels are very similar until a new period of change occurs from the end of 2011 until 2013 when the lake levels return to the pre-2004 state. Compared to the three other study sites, the White Nile watershed and more specifically Lake

Victoria is strongly affected by climate variables due to the predominance of its surface on the basin drainage area, lake surface area represents approximately 42 % of its drainage area. Added to this is a strong anthropization of the outflows which can strongly affect the lake levels. It was therefore decided to focus the analysis on the period before 2004 when the lake is less impacted by the operating rules of its outlet. Even if the outflows are regulated, simulations exhibit good performances in terms of retrieving both the timing and the magnitude of the lake levels before 2004. Moreover, the high water levels in 1998 are well simulated with a peak discharge which is well represented (Fig. 9). The standard deviation over this period shows good results with an $\alpha = 1.1$ ($\overline{\sigma}_s = 0.37$ m, $\sigma_o = 0.35$ m). In addition, the correlation is very good over this period with a score of 0.83, while the RMSD stays low with an average of 0.36 m. The Taylor diagram for Lake Victoria exhibits a best performance of the $ctrip\_mlake\_w1$ to retrieve the pre-2004 lake levels. Both a larger or a smaller width deteriorates the correlation and increases the variability of the levels. However these impact are quite small, and the results on the pre-2004 period still gives acceptable scores.

The seasonality of the simulated lake levels shows a good agreement with observed levels in accordance with a relatively good correlation (shown in Fig. 11). Lake Geneva is the only lake which exhibits low quality level variations in contrast to Lake Baikal and Lake Ladoga which, despite a temporal shift of approximately two months mainly for low flow periods, shows strong correlation for the seasonal pattern. On these lakes, the model simulated the winter low flows well which were linked to soil freezing and low solar radiation (thus little to no melt), but also to the spring high water period resulting from snowmelt. The strong decrease in Lake Victoria water levels during the period 2004-2006 does not significantly affect the climatological cycle which shows good agreement on the seasonal pattern in terms of representing the wet season high water levels and low flows occurring in October.

At every study site, $ctrip\_mlake\_w5$ remains the configuration with the lowest scores which is mainly caused by higher water releases resulting in lower water level variations. On the other hand, both $ctrip\_mlake\_w1$ and $ctrip\_mlake\_w05$ show better agreement to capture the natural variations of lake levels even if local discrepancies, for example inability to capture high water levels on Lake Geneva or temporal shift for Lake Ladoga, appear.

## 5.3 Impact on river discharge simulations

The simulated daily river discharges for the four study sites are shown on the hydrograph in Fig. 12. Even though the Rhône network resolves five lakes at 1/12° resolution, the river flow is mainly impacted by Lake Geneva at the border between Switzerland and France. The lake outlet is located in Geneva in the upper part of the watershed where the water dynamics are lead by a continental snow-dominated climate. The flow follows a bi-modal pattern with low flows in summer and winter while peak discharge generally occurs in Spring. The Rhône basin is the smallest watershed in this study and, due to the importance of the karstic aquifer on the flow regulations, CTRIP has been used with the groundwater options activated. Therefore the selected gauge station is located at the outlet of the basin in Beaucaire (France: Fig. 7). The Beaucaire station is representative of the total Rhône drainage area which also includes the the Mediterranean region which is characterized by intense autumn runoff

associated with strong storm events. As shown in Section 5.1, simulations show significant improvements in the timing and the amplitude of the Rhône discharge at the different study stations owing to the inclusion of lakes. CTRIP simulations are more in line with observations when the lakes are included which consequently improve all metrics in the watershed. The variability is reduced to a magnitude that fits the observed river discharge ($\overline{\sigma_s} = 1064$ m$^3$s$^{-1}$, $\sigma_o = 1003$ m$^3$s$^{-1}$). The important flood events in autumn 2002 are well represented by $ctrip\_mlake\_w1$ with good results for the flood pattern and variability. In particular, the model captures well the consecutive flash floods during autumn 2002 with a well-produced alternation between high discharges and low flows.

Improvements of the NSE and KGE are particularly high over the Rhône river (Table 6). Looking at the distribution of scores along the network, Nash-Sutcliffe scores are higher downstream compared to those at the lake outlet. Compared to the reference simulation $ctrip\_nolake$, NSE scores increase by 19 % at Beaucaire ($\overline{\text{NSE}} = 0.69$) while NSE_log increases by 88 % ($\overline{\text{NSE}_{log}} = 0.64$). Lakes introduce a better representation of extreme events on the Rhône river with even better improvements on sustaining low flows. The KGE score is more influenced by bias and variability than the NSE, which is weighted more by the correlation scores. Over the Rhône river, KGE scores are slightly improved (by 13 % at Beaucaire: $\overline{\text{KGE}} = 0.85$). Both local and regional streamflow variability and magnitude are better when taking lakes into account with a slight tendency to over-estimate low flows. The NIC score has been calculated using the Nash-Sutcliffe coefficient in order to quantify the contribution of the lake model compared to the baseline scenario. It reveals a mean improvement of 25 % of the NSE scores at Beaucaire which further corroborates the positive effect of the inclusion of lakes dynamics. The lake outlet width which is half of the initial value leads to better results for every metric. The Rhône streamflows are globally improved with the magnitude depending on the location within the network. However, the high frequency dynamics at the lake outlet are not captured in any configuration. In terms of variability, lakes impact the number of peak discharge events and the volume of water transferred during these events. The hydrographs are then smoothed owing to the damping effects of lakes. This is reflected in the seasonal cycle with snowmelt occurring in spring with the greatest streamflows during winter associated with low flows due to mass retention (by the snowpack) in the upstream area.

For the other catchments, the introduction of lakes has a rather small impact on the scores. The main improvements resulting from the inclusion of lakes is a better representation of variability. The analysis of the White Nile simulation is constrained by the discharge measurement availability. Lake Victoria is a buffer for watershed flows and its outflows follows the same pattern as the atmospheric forcings with a succession of low flows during the dry season and peak discharge during the wet season. Despite the Agreed Curve and the improvements resulting by including the lake model, CTRIP-MLake simulations do not capture well the peak discharge. However, the seasonality is well captured with the succession of increasing discharges during wet season and decreasing discharges during dry season. The effect of lakes is consistent with the Rhône results, but daily discharges are slightly under-estimated. Lake Victoria acts as a large retention area which sharply reduces and delays discharge peaks. Evaluation outflows data for the White Nile are available only on the period 1983-2006, owing these limited data, the metrics have computed on this period in this particular case. The average discharge is in the same order of magnitude compared

to observations with an average under-estimation of 2.7 % ($\overline{Q}_{ctrip\_mlake}$ = 1028 m$^3$s$^{-1}$, $\overline{Q}_o$ = 1057 m$^3$s$^{-1}$). The main lake effect is on the standard deviation which decreases by 55 % ($\sigma_{ctrip\_mlake}$ = 736 m$^3$s$^{-1}$, $\sigma_O$ = 300 m$^3$s$^{-1}$) which indicated an improvement of simulated discharge variability. Regarding the specific period 2004-2006 which corresponds to the large lake level decline, there are no results showing a sharp decrease of the outflows that would result in a specific runoff reduction or evaporation increase. However, the measured outflow seems to have reduced variability while tending to increase slightly from 2000 to 2005 (further discussed in Section 6.2. There is no direct result on the simulated outflows that can explain the observed dynamic. Over the 1983-2006 period, NSE are negative for Lake Victoria outflows which confirm the rather small effect of lakes in simulating lake outflows (Table 6). However, NSE and KGE scores are not worsened but remain very low and reveal an inadequacy of the model to retrieve White Nile discharge at the outlet of the lake ($\overline{NSE}$ = -17.6, $\overline{NSE_{log}}$ = -8.3, $\overline{KGE}$ = -2.9). The NIC scores show an average improvement of 78 % of the NSE due to the inclusion of lakes. $ctrip\_mlake\_w05$ is the configuration that gives the best results with an improvements of 93 % and a deviation ratio of 1.54. Even though there is room for improvement, the representation of Lake Victoria has a significant impact on the White Nile streamflows compared to the reference ISBA-CTRIP simulations.

The Angara basin is dominated by Lake Baikal which is the world's largest freshwater continental reservoir. This river is also anthropized with three large dams and its outflows are, thus, regulated by man operating rules. Fig. 12 shows the relatively poor performance of the non-calibrated lake module to retrieve anthropized streamflows and only $ctrip\_mlake\_w05$ is producing a positive NSE value (Table 6). Even though the daily discharges are not well captured, simulated and observed streamflows show good agreement in terms of seasonality with peak discharges and low flows which are reasonably well represented in time and magnitude. $ctrip\_mlake\_w1$ captures the flow variability which is confirmed by a variability ratio of 1.02. With the exception of $ctrip\_mlake\_w5$, all of the configurations significantly improve the streamflows simulations. This reduction of the peak discharge originates from the large retention capacity of Lake Baikal. With a volume of 23,260 km$^3$, this lake has a water residence time of approximately 330 years which substantially affects the regional hydrology. The average discharge is generally well estimated ($\overline{Q}_{ctrip\_mlake}$ = 1851 m$^3$s$^{-1}$, $\overline{Q}_{Obs}$ = 1860 m$^3$s$^{-1}$) with a difference of only 0.5 % between simulated and observed average discharges. As was the case for the other lakes, the explicit modeling of Lake Baikal sharply reduces the standard deviation by ($\sigma_{ctrip\_mlake}$ = 625 m$^3$s$^{-1}$, $\sigma_{Obs}$ = 486 m$^3$s$^{-1}$) related to it particularly high buffer effect on the catchment hydrology. Performance scores over the period 1983-2013 are improved by the integration of lakes with a stronger effect on the NSE which increases by 0.19 for $ctrip\_mlake\_w05$. More generally in terms of NSE skill, all lake configurations improve the scores but only $ctrip\_mlake\_w05$ gives a positive score (NSE =0.26) which is due to a better simulation of low flows (NSE$_{log}$ = 0.23). Even if the NSE is low, the NIC score shows an average improvement of 87 % (ranging from 0.75 to 0.94) owing to the introduction of the lake model. Lake Baikal is covered by ice generally from January to May-June, and it is surrounded by permafrost. This specific seasonal process is the main driver of the regional hydrological pattern with low flows during winter and high peak discharge caused by snowmelt during the Summer. Since the Nash-Sutcliffe score is quite sensitive to flow peaks (making it especially sensitive to the seasonal snowmelt runoff in this basin), it makes more sense to limit the result to both the KGE and NIC performance. Thus, even if all lake configurations seem to improve streamflow simulations,

$ctrip\_mlake\_w05$ produces the best results.

The Neva river takes its origin in Lake Ladoga which is itself fed by the Onega lake. This region is of particular interest owing to the high lake density which strongly affects the streamflow dynamic. The inclusion of lakes in Scandinavia reveals a significant impact on streamflows as shown in Section 5.1. All lake configurations significantly improve the results in terms of volume transferred with a significant decrease in the peak discharge. However, the hydrograph of the $ctrip\_mlake\_w5$ is

660 characterized by an over-estimation of the peak discharges which is confirmed by a variability ratio of 1.47. On the other hand, the $ctrip\_mlake\_w05$ simulation tends to under-estimate the flow variability ($\alpha$ = 0.65). The average discharge generally captures the observed flows with a difference of only 2 % ($\overline{Q}_{ctrip\_mlake}$ = 2538 m$^3$s$^{-1}$, $\overline{Q}_{Obs}$ = 2485 m$^3$s$^{-1}$). Adding lakes generally improved the simulated variability ($\sigma_{ctrip\_mlake}$ = 634 m$^3$s$^{-1}$, $\sigma_{Obs}$ = 655 m$^3$s$^{-1}$) with strong improvements for $ctrip\_mlake\_w1$ simulations (the variability ratio is 0.92). However, the hydrographs show a systematic time shift between

665 the simulated and the observed daily discharges (Fig. 13). This shift has significant consequences on the performance metrics by deteriorating the correlation between simulated and observed discharges despite the overall reasonable fit to the hydrograph. As shown in the Table 6, NSE scores are negative in two configurations ($\overline{NSE}$ = -0.71) and positive for $ctrip\_mlake\_w05$ with a strong improvement (NSE = 0.26). In contrast, the KGE score, which reduces the weight of correlation, shows a high score for $ctrip\_mlake\_w05$ and $ctrip\_mlake\_w1$ simulations ($\overline{KGE}$ = 0.19). The positive effect of including lakes

on this Scandinavian region is supported by the NIC score (which evaluates the improvement brought by the lake module) which increased by 81 % for $ctrip\_mlake\_w05$. However, it worsened the streamflow simulations in the $ctrip\_mlake\_w5$ configuration (NIC = 0.15).

Seasonal discharge is presented in Fig. 13. Generally speaking, the introduction of MLake allows a better representation of the seasonal cycle of the river discharge for the four study sites. However, these improvements are heterogeneous along

the basins. In terms of variability, the impact of including lakes on the seasonal cycle leads to the reduction of both the mean discharge and the temporal variability. For the Rhône basin, the introduction of lakes increased the low flows simulations and reduced the river peak discharges. The best results are for Lake Baikal where the river discharges are sharply reduced and are much closer to those observed. The main result of the analysis of the seasonal cycle is the difficulty of $ctrip\_mlake\_w5$ to simulate well the river discharge, with the exception of Lake Geneva. However, it is not possible to point out if the main reason

is strictly coming from the sensitivity to the weir width or another process that is not yet represented such as reservoirs water management.

## 6 Discussion

### 6.1 Lake internal dynamics

Simulations reveal the capability of the non-calibrated CTRIP-MLake system to capture lake level variations and to improve the

685 simulated river discharge. However, it is important to note that, despite the explicit representation of some spatially-distributed processes within the model such as runoff, the model resolves an one-dimensional water balance equation. This means lakes,

regardless of their size, are represented as points in the network. This representation could be problematic for large lakes, such as Lake Baikal, where wind-stress effects on the lake height or internal wave processes (which are not included in the model) could impact the overall lake dynamics. It also means the diagnostic variables are redistributed over the lake network mask and affect the regional performance by introducing local biases. Observed height differences over lakes can reach several meters from one shore to another depending on the wind stress and the distance of the fetch among other factors, and consequently this can influence the relatively high frequency variability of river discharge. In that sense, local and regional assessment would benefit from developing a specific diagnostic computation applied for large lakes. This specific diagnostic could, for example, consider level differences from one shore to another in a simple way without the need to introduce hydrodynamic processes. One of the easiest approaches could be to also take into account simple bathymetry in order to characterise a distributed water layer. Modelling could also benefit from observations datasets. As was done for lake Geneva, these gaps could be overcome by gathering data from several measurement sites along the lake shore, but this depends on the data availability. Over the long-term, comparison between modeled and observed water levels could be improved by valuable satellite data as proposed in the Surface Water and Ocean Topography (SWOT, Biancamaria et al., 2016). In this context, sub-grid variability could be improved by considering larger lake as a mesh where each grid cells could interact with each other. However, within the scope of the current study, the inherent model gaps have relatively few impacts on lakes with a small surface areas or on regulated lakes. Therefore, long temporal series, such as those which are characteristic of climate studies, render these processes negligible. Daily comparison must be cautiously made and long-term, seasonal analysis must be prioritised. MLake is intended for use in long-term monitoring of large scale basins with respect to the inherent framework of global scale climate studies. In addition, the use of a non-calibrated model restricts the performance for local daily evaluation as lakes are not impacted by rules based on observations that minimize errors through the modification of adaptable parameters.

## 6.2 Lake anthropization

### General impacts

Most of the world's rivers are anthropized leading to an additional number of factors that modify the natural land surface process variability and hydrological variables such as runoff and streamflow (Grill et al., 2019; Best, 2019). Lakes are no exception to this evolution and the lake level fluctuations seen in the current study are consistent with those resulting from worldwide anthropogenic regulation. Regarding the impact on the daily outflows, Lake Baikal seems to have less natural outflows since observed discharge is characterized high variability resulting from a strong anthropization which is impossible to simulate here. This pattern is mainly introduced by the joint effect of the three reservoirs constructed on the downstream river (Irkoutsk, Bratsk and Ust'-Illim reservoir). Beyond a simple local effect, these reservoirs affect the regional streamflows with a signal seen as far as on the Yenisei river (Adam et al., 2007). Among these anthropogenic structures, the Irkoutsk dam, which is the principal regulator of the Angara dam chain. Located just 55 km away from the Baikal outlet, it has increased the lake storage capacity by 37 km$^3$. One of the main objectives of this dam is to restrict outflows in order to limit peak discharge and to sustain baseflow while trying to keep a natural cycle characterized by high levels in autumn and low levels at the end of winter. In the

720 dry season, the regulation sustains river baseflow by decreasing of the outflow. During the wet season, regulations control the outflow in order to refill the lake. This specific variability can explain the fairly poor performance of NSE scores (which is significantly reduced by low correlation values). MLake currently does not take any operating rules into account and is intended to only retrieve natural streamflows produced by the hydrological components. The direct consequence of the Angara regulation is the increased river discharge associated with a decrease of the maximal variability by one third that leads to a significant rise

of the lake water level (Vyruchalkina, 2004). However, processes that allow water abstraction, include downstream irrigation demand or represent dam operating rules are not modelled which explains most of the discrepancies in both lake level and streamflow estimations.

Similar effects occur for Lake Geneva where the Seujet dam regulates the outflows and thus the lake levels. The dam effect on the seasonal cycle is particularly clear where a cut-off the high water levels occurs in Spring resulting in an absence of peak

discharge for the Rhône river. On the other hand, during the summer, the dam helps to sustain baseflows thereby mitigating the impact of droughts in the basin. An inter-comparison of the different study sites reveals that Lake Ladoga is the only lake that is not actually regulated. Neva is flowing on a natural riverbed which includes the gauge stations locations just before the river reaches Saint Petersburg. Despite the temporal shift inherent to CTRIP physical processes (discussed later in Section 6.3), the model evaluation using observations shows good agreement. Anthropogenic impacts on lake levels are reduced on larger lakes

since the ratio between regulation dynamics and the lake temporal response to the natural forcing are not in the same order of magnitude. The upstream influence of dams on river discharge does not significantly impact the performance skill.

**Closer look on the Lake Victoria historical level drops**

Anthropization has a clear impact on Lake Victoria and can explain many of the discrepancies between model simulations and

740 observations in the current study. The unique gauge station with continuous data is located on the Nalubaale dam complex in Jinja (Uganda) just a few kilometers downstream of the Lake Victoria outlet. In 2000, a second dam was commissioned on the White Nile river at Jinja called the Owen Fall complex. Outflow from the complex is administered under an agreement called the "Agreed Curve" which restricts the outflow rate in order to mimic natural lake outflow. This makes it rather difficult to assess the impact of such anthropization downstream and to temper the conclusion and scores. Several studies have attributed

the severe 2004-2006 Lake Victoria level decline to both historical regional drought and the impact of outflow deregulation (Kull, 2006; Sutcliffe and Petersen, 2007; Vanderkelen et al., 2018; Getirana et al., 2020). According to these studies, half of the decline could be the contribution of the dam over-release and the other half could be attributed to a severe drop of the runoff resulting in very low inflows in the lake. More specifically, Vanderkelen et al. (2018) determined that the PERSIANN-CDR precipitation estimation for the watershed decreased by 13 % compared to the time averaged precipitation over the period

2004-2005 with an impact on the levels of all surface water in the region. However, the reduction in the forcings can not explain the drop in lake water levels of 1.19 m (equivalent to a volume of 83 km$^3$).

The verification of the precipitation forcings and the assessment of the regional drought signals reveals an over-lake runoff anomaly of -0.20 mm for the hydrological year 2004-2005. The associated mean lake levels drop is 0.39 m (interval of [0.25 m-0.57 m]) which is not enough to explain the observed value of -1.04 m. Another important driver that affects the lake water balance is the outflow attributed to the dam complex. Getirana et al. (2020) showed a strong effect of the non-respect of the Agreed Curve with a sharp increase of the dam releases during this time period. Sutcliffe and Petersen (2007) found out that the additional level drop is about 0.61 m and it was caused directly by the over-abstraction for the period 2004-2005 in line with the study of Kull (2006) which stated that 55 % of the lake level drop was caused by dam over-release. MLake does not account for dam operating rules and irrigation/abstraction which explains the emergence of the gap between the simulated and the observed levels. However, the model simulates a decline of the lake levels that can be attributed to the runoff decrease in line with studies showing that the lake level response to the droughts would have been by approximately 0.3 m (Vanderkelen et al., 2018). Lake Victoria is not the only lake affected by anthropization, and some studies have already tried to prvide warnings on the current stress that humans are imposing on the lake levels and the water balance around the world (Wurtsbaugh et al., 2017; Jenny et al., 2020).

## 6.3   Temporal shifts on simulated Boreal river discharges

Shifts between observed and simulated streamflows and levels are not only caused by anthropisation and forcing input, but obviously they also arise from inherent gaps in the ISBA-CTRIP model. Results for the Lake Ladoga outflow, which is relatively free of anthropogenic effects, show a systematic two-month temporal shift of early peak discharges compared to observations. As discussed in Decharme et al. (2019), simulations of river discharges north of 50° N latitude are constrained by the ability of the model to reproduce snow melt. The version of ISBA used within the current study (which has been the historical default scheme used in ISBA-CTRIP) solves a unique composite energy budget for the soil and vegetation and therefore does not account for the radiative effect of the forest on the underlying snow. These gaps can be particularly important in Boreal forest zones. The neglect of these processes generally causes an early peak in snow melt runoff and therefore river discharge. The use of the ISBA Multi-Energy-Budget scheme (Boone et al., 2017) could lead to improvements in estimations by attributing an independent energy budget to the vegetation, the snow and the soil. It also includes specific processes such as interception and unloading of the snow by the canopy. A recent study shows how this model improves the timing of snow melt timing at Boreal forest sites for time scales which are consistent with the errors identified in the current study (Napoly et al., 2020).

## 6.4   Simulation sensitivity to the lake outlet width

A simple sensitivity analysis has been performed in order to provide a broad evaluation on the model response to the width of the lake outlet. In general, the hydrological statistical evaluation metrics are improved with a reduced width and worsened with an increased width (above the initial baseline value). However, there are significant improvements in the response of the lake outflows during extreme flows when the weir width is larger. In these specific cases, reducing the weir width will increase the flood-wave buffer property of the lake. As explained in Decharme et al. (2019), river widths in CTRIP are computed based on the annual mean discharge. The weir width corresponds to a fraction of the total lake circumference. In the model, lake

morphometry is reduced to an equivalent lake circle for which the circumference is smaller than that of an equivalent (size) natural lake. This widening produces an overestimation of the outflows which can explain, in part, the better performance of smaller weir width simulations (such as the simulation labelled $ctrip\_mlake\_w0.5$ in Section 5.3). Global datasets with information on lake morphometry remain scarce and require extensive human intervention and financial resources. Furthermore, it might be impossible to gather reliable information on such on unsteady parameter which depends on either the downstream river morphology and the actual water head over the crest. A possible improvement would be the estimation of a generic width as a fraction of the lake circumference by clustering lakes based on both the observed outflows and the morphometry type.

The characterization of the lake morphometry is one of the main sources of uncertainty in such models. First, natural lake outlet widths are not constant and are generally a function of the lake level which induces potential outflow overestimation during low flows. The diagnostic level inferred in the model depend on the assumption of a linear lake hypsographic curve leading to the representation of the lake bathymetry as a cylinder. Bathymetry is of particular importance for lakes as it controls most of the inherent biological, physical and ecological processes (Blais and Kalff, 1995; Håkanson, 2005; Yao et al., 2018). In the current study, lake bathymetry influences the residence time and magnitude of variations in both levels and surface area. This limitation will be addressed through the integration of a specific global scale hypsographic curve that can fit most of the lakes morphometry. In order to satisfy global scale properties, the lake morphometry computation must remain computationally efficient and thus should be based on a relatively simple hypothesis that can capture the wide diversity. To do so, a specific study should be done to create a global scale dynamic data set in the SURFEX platform to account for semi-permanent areas surrounding lakes. Further developments will also require the integration of a strategy to correct the land cover type of these flooded areas during the dry season. The dynamic will allow the introduction of wetlands that would interact through sub-surface fluxes with the lake and permit an improved estimation of the evaporation.

**Coupling MLake to the SURFEX modelling platform**

Among all the further developments, the effective coupling between MLake to the SURFEX platform will help many features to be improved. Evaporation estimations will gain in accuracy with a fully coupled MLake-FLake system that will simulate the feedbacks of the lake within the global hydrological cycle. Furthermore, the coupling will also improve the simulation of lake surface freezing which remains one of the major limitations that could influence MLake. In the current version, only Flake explicitly represents frozen lakes in the energy budget.

At the moment, CTRIP is coupled with SURFEX through the ISBA model but MLake is only available for off-line simulations. In the future configuration, MLake will provide the diagnostic variables representig lake level and surface area that will be used by MLake as part of the latent heat fluxes computations.

## 7  Conclusions

Hydrological and meteorological developments based on global scale models are at the core of the CNRM research. Recent changes allow an increased number of processes to be integrated into both climate (CNRM-CM6) and hydrological models (ISBA-CTRIP). Following the recent updates of groundwater and floodplain processes in the land surface and hydrological model ISBA-CTRIP, the purpose of this study was to evaluate the performance of the inclusion of a non-calibrated mass-balance lake model in the modelled water cycle. This offline evaluation was conducted over four river basins using a unique validated atmospheric forcing dataset and a combination of both in situ and satellite measurements for river discharge and lake level observations.

Even if the main responses of ISBA-CTRIP to the inclusion of the new lake model are different among the selected test basins, several key improvements to the model simulations were identified. The addition of lakes in the river network reduced the average variability of river discharge by 34 % which lead to a lower number of simulated peak discharges and improved baseflow compared to observations. The Kling-Gupta Efficiency score is not improved to the same degree for all four studied basins. However, improvements are notable on all basins except the White Nile basin. These improved performance was size dependent and the introduction of larger lakes drastically improved the streamflow simulation metrics. The average KGE score for Lake Geneva was 0.81 while it was 0.18 for Lake Baikal. Note that the NIC score has also been estimated which is more appropriate for assessing the actual improvements owing to the introduction of MLake. The mean NIC score for this study is 0.57 compared to the reference run (interval of [0.15-0.93]). The NIC scores are very sensitive to the chosen lake and improvements are notable for Lake Baikal with a mean score of 0.87. The inter-annual variability was improved even if some discrepancies appear such a persistent over-estimation of the mass transferred affecting low flow simulations: the average ratio of river discharge was 0.998 (interval of [0.86-1.45]) with a particularly good simulation of average discharge of the Angara river (a mean ratio of 0.99). Moreover, the model did not correct the early peak discharge in the boreal zone coming from precocious snowmelt, but new model physics to be introduced into ISBA-CTRIP should improve this. It is also important to point out that simple assumptions made for retrieving natural outflows imply that the model is unable to retrieve observed anthropized high frequency dynamics. Regarding lake levels variations, MLake is capable of simulating realistic lake dynamics in all study sites with a mean correlation of 0.56 (ranging from an average of 0.28 for Lake Geneva to an average of 0.83 for Lake Victoria). These results are particularly encouraging for Lake Victoria since droughts are well represented compared to the observations.

The new model parameter is the lake outlet width and an improved method to monitor or estimate its width could increase significantly the hydrological statistical performance metrics. For the four lakes studied, the simulation with an initial weir width divided by a factor two showed the best scores for both river discharge and lake level simulations. Furthermore, the introduction of lakes in the river network is of particular importance in terms of global water flux as the lake water dynamics not only have an effect on the local hydrology but also the streamflow at the outlet of the basin. Last but not least, the most important characteristic of the model is it's ability to improve the seasonal cycle for both lake level and river discharge for every

study site. All this advocates for results to be extended to the global scale in order to characterise the systemic improvement for an ensemble of climate and physiographic conditions.

Lake dynamics are sensitive to external stresses such as anthropization (dam operating rules, irrigation extraction or water abstraction) which limits the capabilities of the model to retrieve observed discharges for many basins. The lake impacts are heterogeneous in time and space among the watersheds and limit the possibility of capturing a specific and systematic pattern on streamflows. The performance can not be improved in such cases without degrading simulations elsewhere and a specific reservoir model is necessary to correct streamflows locally. Numerous studies have been focused on such developments (Hanasaki et al., 2006; Zhou et al., 2016; Busker et al., 2019; Shin et al., 2019) and on-going research is focusing on creating a global reservoir system that will be added to MLake to improve the representation of dam operating rules. Simulations were also constrained by the off-line use of the energy budget model FLake which computes over-lake evaporation, and further improvements to the model system will be made by coupling MLake to FLake (meaning extracting evaporation computed by FLake from MLake as a first step). In order to propose a fully coupled dynamic model and to take riparian land cover changes into account, the introduction of dynamic cover maps in the SURFEX modelling platform is necessary. This integration of a dynamic cover fraction will replace a proportion of land covered by lakes during periods of high levels, and, conversely, it will lead to an increase of the land fraction during low levels. Thus, it is a first step towards implementing semi-permanent waters as proposed in the map by Pekel et al. (2016). Currently, the main goal of the developments described in this study are to fully couple MLake within the SURFEX platform for improved representation of lake dynamics in global scale hydrological and climate studies. This updated SURFEX-CTRIP-MLake system will help in different domains such as drought risk management and water resource management in a context of global water resource scarcity. It will also contribute as an important component of earth system models by permitting a long-term quantitative assessment of the fully-coupled global water cycle, it's trend and its inherent variability. Within this high-resolution river routing, it is not just a new stage that has been reached in global scale hydrology but also the upper limit of model without considering hydrodynamic processes. Improving the resolution would lead to the inclusion of hydrodynamics processes such as currents and internal waves but would also need a discretization of the sections of the largest rivers.

*Code and data availability.* The SURFEX v8.1 model platform code, which contains the ISBA and FLake codes, is available in the supplement. All post-processing codes are also available. Finally, model output for all of the study sites and forcing data are available in the supplement. The supplement is available at: https://doi.org/10.5281/zenodo.4013873

## Appendix A: Performance skill

### A1 Discharges evaluation

The simulated river streamflows were evaluated using a set of statistical metrics that are widely used in hydrological modelling. The dimensionless Nash-Sutcliffe efficiency score (NSE) ranges over the interval $[-\infty;1]$. It assesses the performance of a hydrological model by comparing the simulated discharges to a simple model composed of the time-averaged observations. Positive values provide information on the ability of the model to retrieve the observed discharge dynamic. A value of one indicates a perfect fit between observations and model simulations and a zero-value indicates that the model is able to produce the average of the observations. It is expressed as

$$\text{NSE} = 1 - \frac{\sum\limits_{t=0}^{n} (q_{s,t} - q_{o,t})^2}{\sum\limits_{t=0}^{n} (q_{o,t} - \overline{q_o})^2} \tag{A1}$$

where $t$ is the time, $n$ the number of values (time steps here), $q_{s,t}$ the simulated river discharge at the time step $t$, $q_{o,t}$ is the observed river discharge at the time step t and $\overline{q_o}$ the time-averaged observed river discharge.

The second metric used is the logarithmic Nash-Sutcliffe efficiency score which gives more weight to a model's ability to retrieve low flows (compared to flood peaks) and determines more accurately the model systemic over-estimation or under-prediction during these periods:

$$\text{NSE}_{log} = 1 - \frac{\sum\limits_{t=0}^{n} \left[\log(q_{s,t}) - \log(q_{o,t})\right]^2}{\sum\limits_{t=0}^{n} \left[\log(q_{o,t}) - \log(\overline{q_o})\right]^2} \tag{A2}$$

Even if very popular, this score is more sensitive to extreme values (e.g. in snowmelt-dominated basins, there can be a relatively high variability in surface runoff with a few large peaks dominating the NSE). In order to prevent these effects, other hydrological metrics are chosen, such as the Kling-Gupta efficiency score (KGE; Gupta et al., 2009) or the modified Kling-Gupta efficiency score (Kling et al., 2012). The evaluation in this study has been performed using the modified KGE which equally weights three components: the linear correlation coefficient, $r$, the variation coefficient ratio, $\gamma$, and the normalized bias of the observed discharges, $\beta$. In contrast to the NSE, it gives more weight to the bias and the variability at the expense of the correlation coefficient. It is expressed as

$$\text{KGE} = 1 - \sqrt{(r-1)^2 + (\beta-1)^2 + (\gamma-1)^2} \tag{A3}$$

where $\gamma = \frac{\sigma_s}{\mu_s} / \frac{\sigma_o}{\mu_o}$. $\sigma_s$ and $\sigma_o$ represent the standard deviation of the simulated and observed river discharges, respectively, and $\mu_s$, $\mu_o$ are the time-averaged simulated and observed river discharges, respectively.

Finally, in order to evaluate the lake model contribution to the model performance the Normalized Information Contribution (NIC; Kumar et al., 2009) was used. This metric provides information on the improvement brought by the considered model relative to the maxim possible score improvement: here the NSE for the reference simulation $ctrip\_nolake$. A positive value gives a measure of the improvement of the model whereas a negative value indicates a degradation of the model performance. The NIC score, in this study, is applied to the NSE as:

$$\text{NIC} = \frac{\text{NSE}_{ctrip\_mlake} - \text{NSE}_{ctrip\_nolake}}{1 - \text{NSE}_{ctrip\_nolake}} \tag{A4}$$

where $\text{NSE}_{ctrip\_mlake}$ is the $NDE$ for the CTRIP-MLake simulations and $\text{NSE}_{ctrip\_nolake}$ is the NSE corresponding to the reference CTRIP_nolake simulations.

## A2  Lake level evaluation metrics

The ability of MLake to reproduce the level variations was assessed using metrics such as the linear correlation coefficient $r$ and the Root Mean Square Deviation (RMSD), which gives an estimate of the quadratic mean of the difference between the predicted and the observed lake level variations:

$$\text{RMSD} = \sqrt{\frac{1}{n} \sum_{t=0}^{n} (H_{s,t} - H_{o,t})^2} \tag{A5}$$

where $t$ is the time, $n$ represents the total number of time steps, $H_{s,t}$ corresponds to the simulated lake level variations at the time $t$ and $H_{o,t}$ represents the observed lake level variations at the time $t$.

## Appendix B:  Algorithm description

Large scale hydrological simulations including lakes are generated over using three steps as presented in the Figure. A1. Among these, two steps are dedicated to rivers-lakes processes and the last one is organised in order to generate the forcing files from the SURFEX platform.

## B1  Preparation of forcings files

Runoff and drainage NetCDF forcing files are generated in offline mode from the land surface model ISBA within the modelling platform SURFEX (Section 2.1). A global FLake simulation allow the inclusion of over-lake evaporation in the forcing data prior to the generation of runoff/drainage. Forcing files are used within the numerical computation process to attribute inflows contribution to the mass balance (Eq. 3 and Eq. 4).

#### Initialisation of lakes

This step consists in an externalised procedure that creates a map containing physiographic information and initialised variables. This particular part is currently written with a mix of Python and Fortran90 (working with the Gfortran and Intel Fortran compilers). Several aspects of this step are related to the Section 2.3.1 and Section 2.3.2. NetCDF files are generated from the integration of lake information, aggregated from the ECOCLIMAP database at 1 km resolution and downscaled to $1/12^o$, for the current CTRIP river routing network. Key parameters and variables generated by this pre-processing step are gathered in the Table 2.

#### CTRIP numerical solution

Global scale NetCDF files containing the physiographic parameters and initial storage variables of the desired configuration are prepared during the initialisation. The numerical solution and the water transfer is fully written in Fortran90 and divided in an initialisation stage which creates a subset, if necessary, of the global maps on the study zone (see Step 2 of the Fig.A1). Then the computation program spreads the runoff/drainage forcing data on the river/lake network, created during the preparatory stage, and routes the water mass based in the sequence number. Diagnostic NetCDF files are written at the input time step (generally daily) and contains two diagnostic data type (the outflow discharge and the level of lakes).

This model tends to be user-friendly with an optimised command interface, allowing users with limited support to operate the code. As soon as the code is compiled, the Fortran option namelist needs to be completed in order to give the desired configuration of the run (input zone, computational time step, diagnostic time step) with the technical support of the SURFEX website (https://www.umr-cnrm.fr/surfex/). Then both the preparatory stage and the master program are run by executing two ready-to-run shell scripts.

*Author contributions.* TG, AB, PLM, SM and BD designed the study and determined the methodology. TG and SM developed the model, and integrated lakes into the CTRIP model. TG performed the analysis and wrote the original draft. MG made accessible the updated version of GLDB and helped for its implementation in ECOCLIMAP. All authors contributed to the editing and review of the paper

*Competing interests.* The authors declare that they have no conflict of interest.

*Acknowledgements.* The authors would like to thank Inne Vanderkelen and Damien Bouffard for providing data needed for the evaluation of this work. The authors would also like to thank Marie Minvielle for operating and maintaining SURFEX and Diane Tzanos for help with ECOCLIMAP.

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

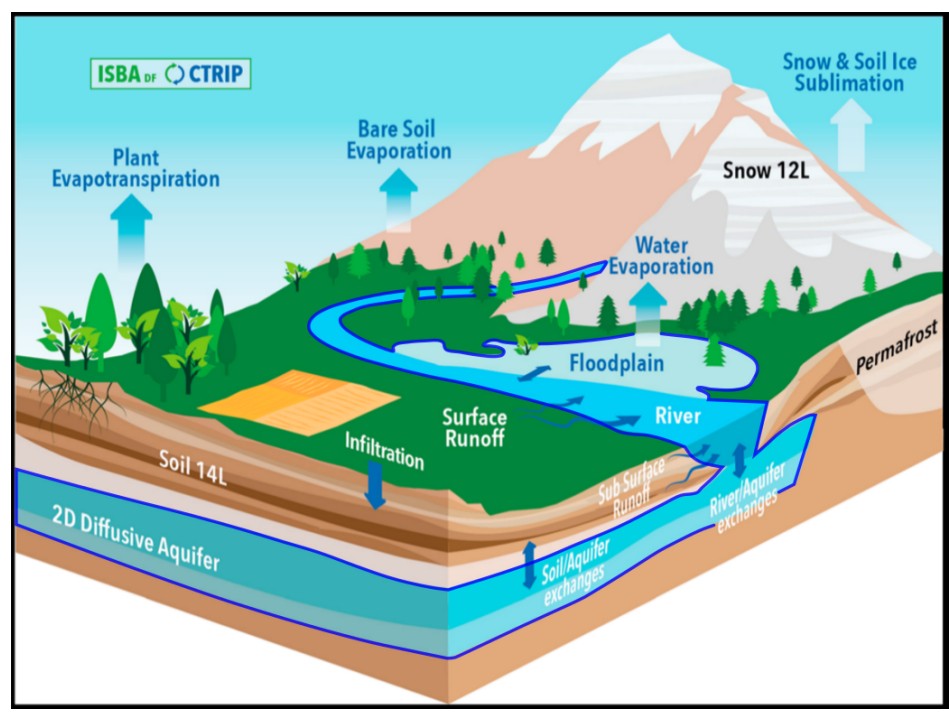

**Figure 1.** Scheme representing the models in the CNRM Climate Model 6 and the processes integrated in CTRIP, adapted from Decharme et al. (2019). The processes represented by the CTRIP model are delimited by the blue domain.

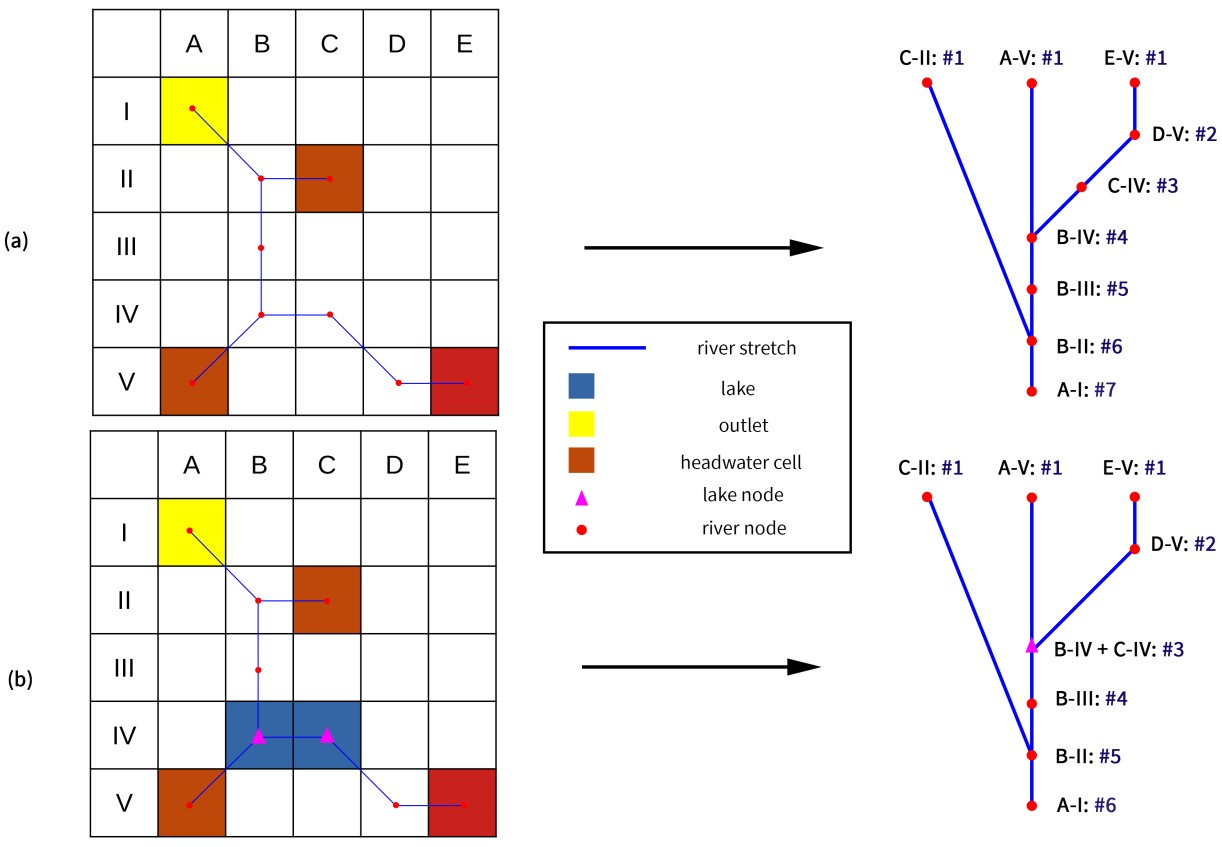

**Figure 2.** Graphical representation of the CTRIP algorithm. a) Spatially-distributed and network representation for CTRIP only. b) The same for CTRIP-MLake.

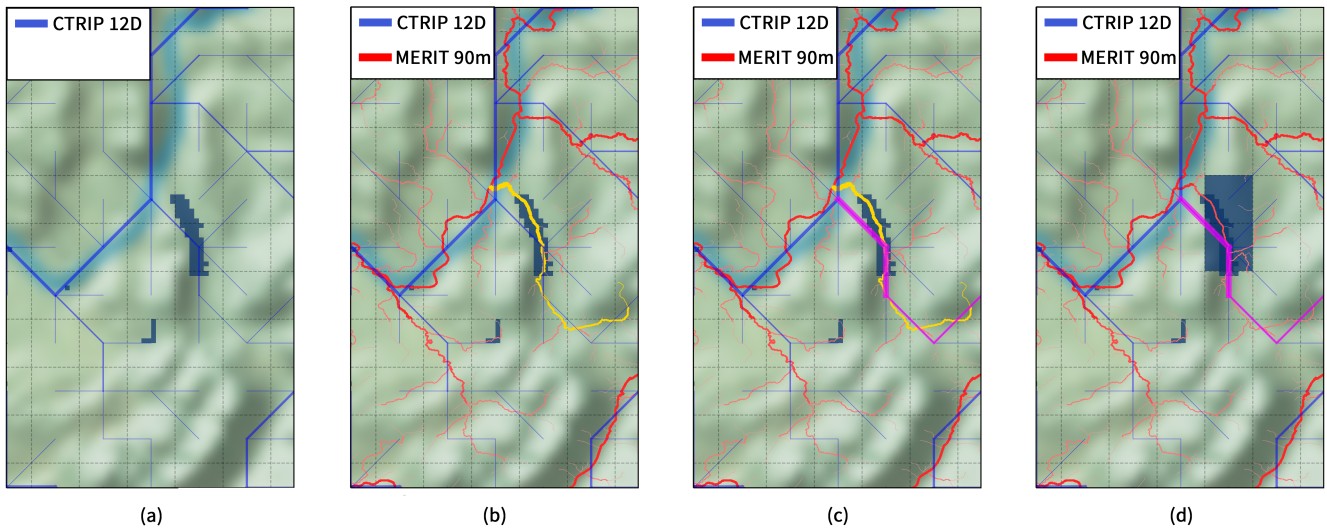

**Figure 3.** Procedure for the integration of a lake in the CTRIP river network at 1/12° resolution. Example for the lake Bourget (France). Fig 3(a) presents the lake Bourget at a 1/120° resolution and the CTRIP river network at a 1/12° resolution. Fig 3(b) shows the identification of the river stretch from the MERIT-HYDRO river network covered by the lake pixels. Fig 3(c) presents the corresponding in the CTRIP 1/12°. Fig 3(d) shows the lake network mask at a 1/12° resolution resulting from the recursive identification using MERIT-HYDRO.

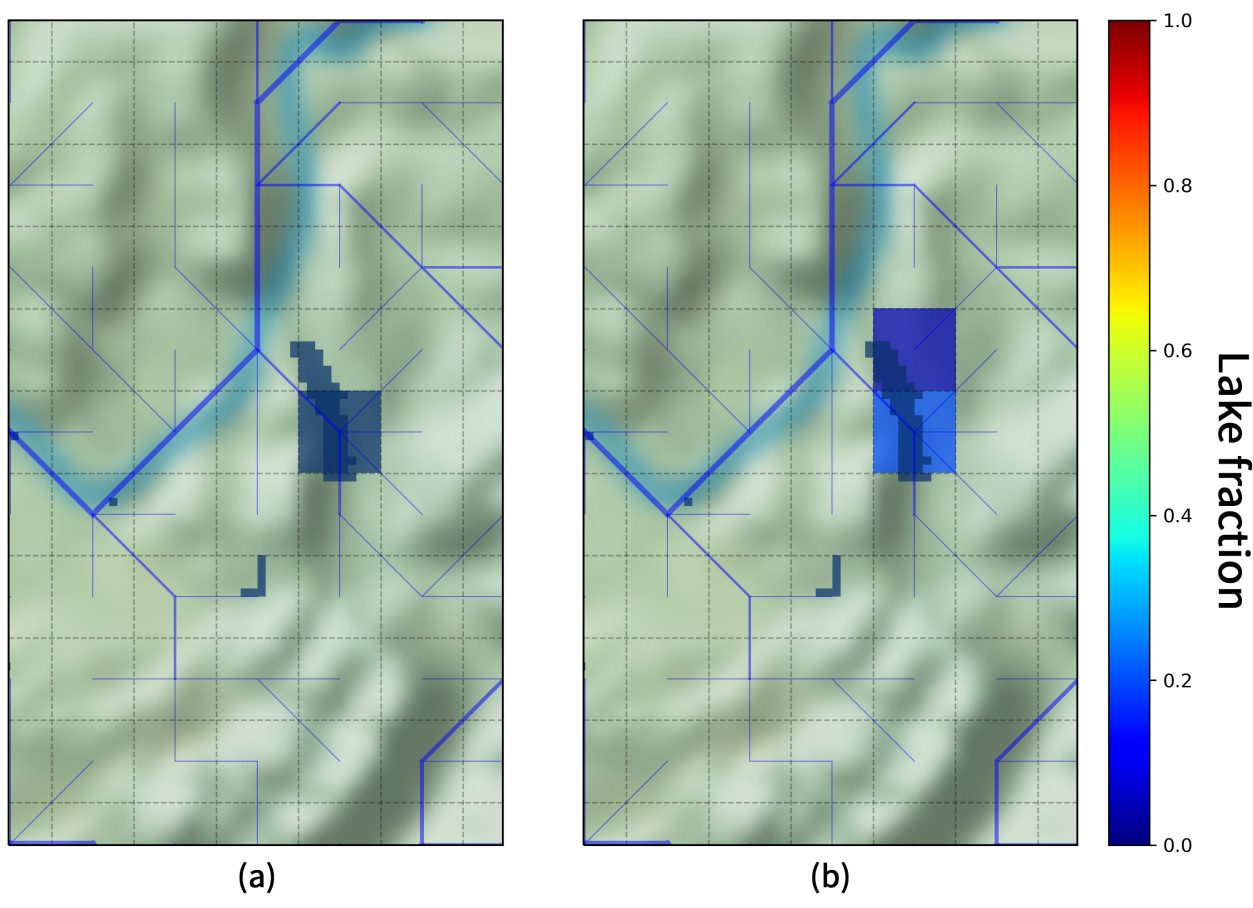

**Figure 4.** Example of a network (a) and runoff (b) masks for the lake Bourget (France)

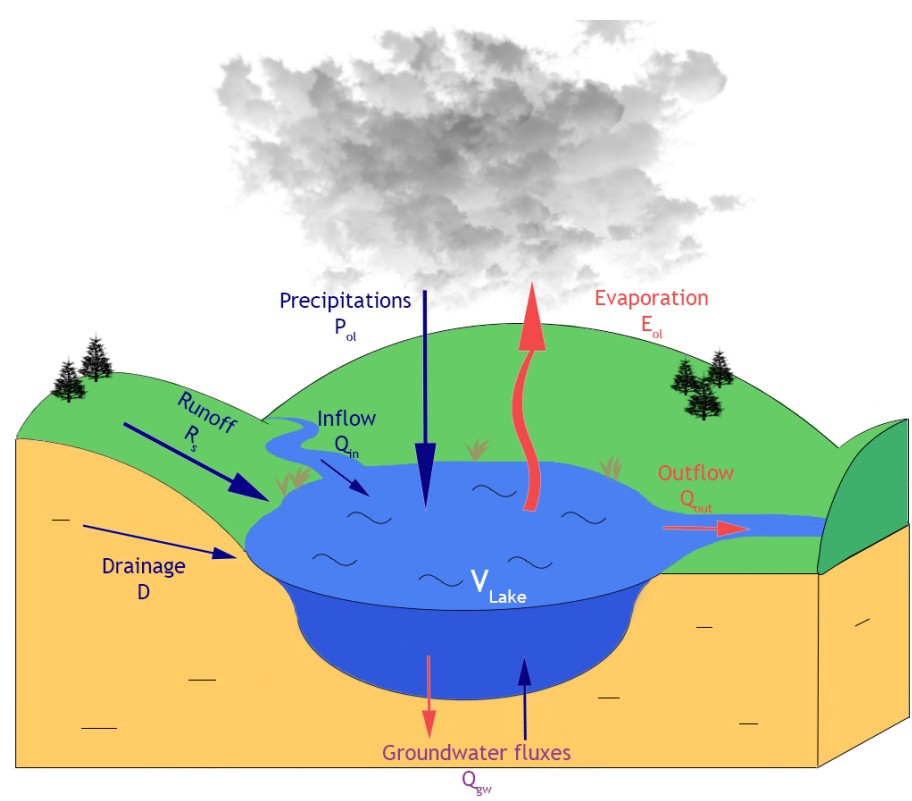

**Figure 5.** Schematic representing the process participating in a lake mass balance evolution

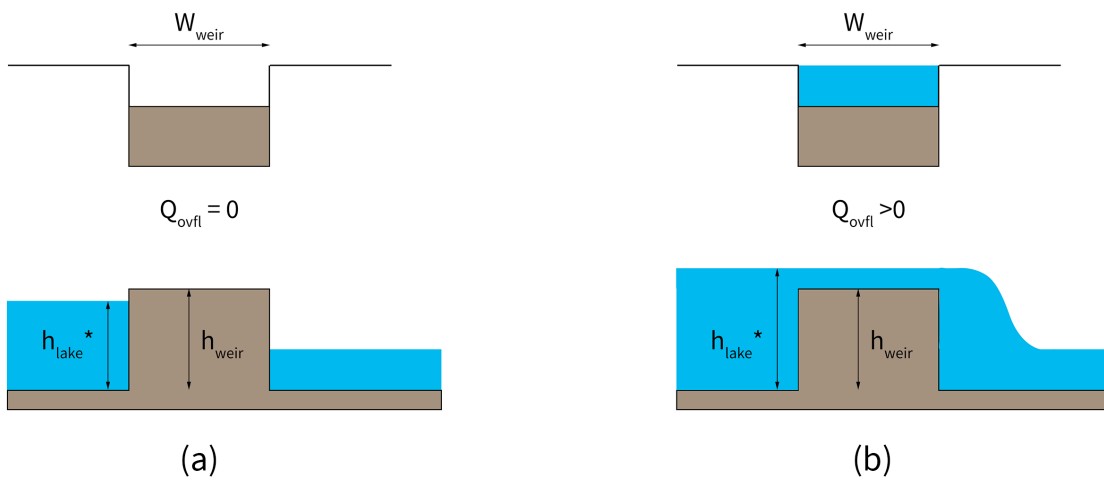

**Figure 6.** Lake-river interaction through overflows

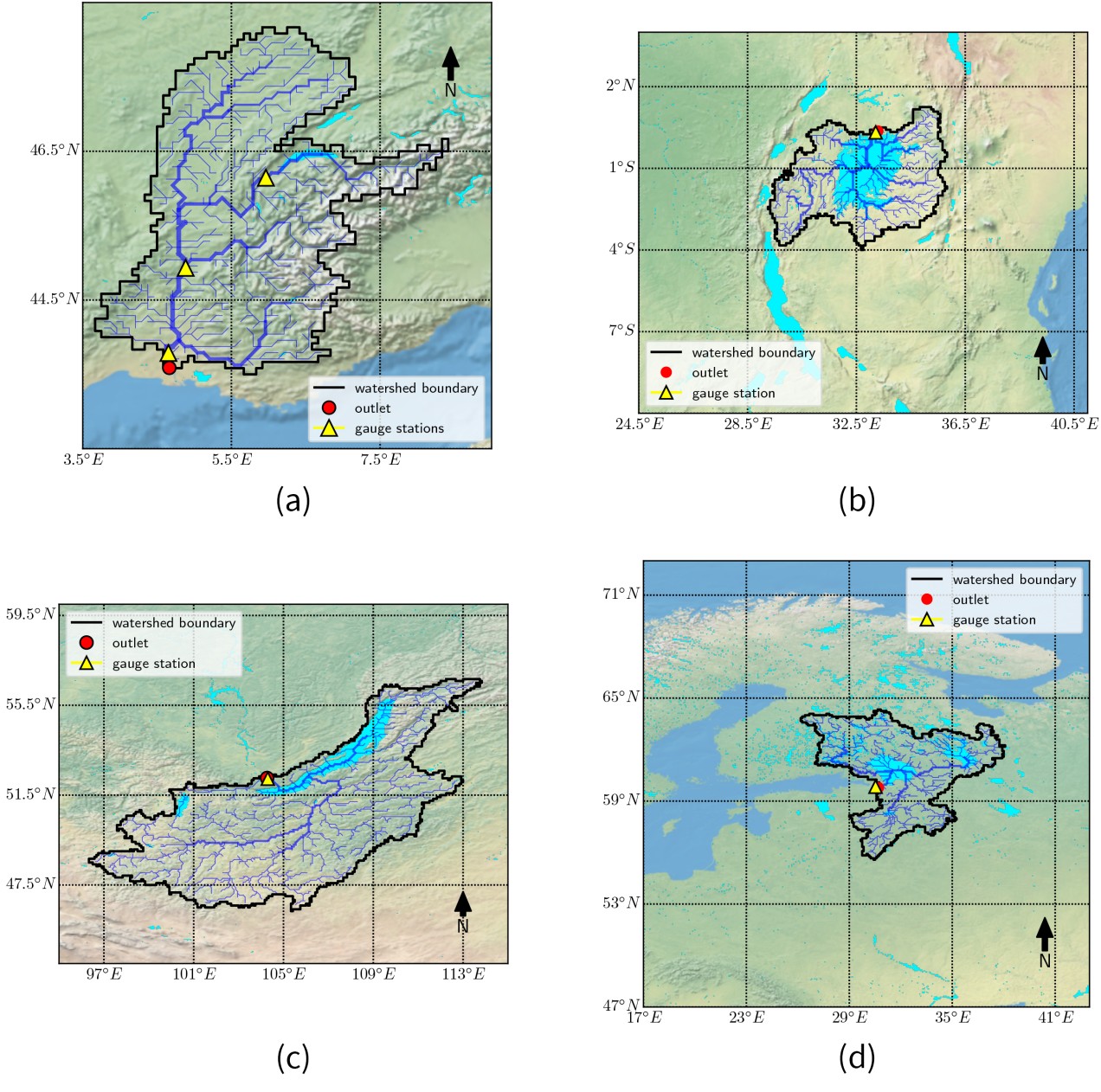

**Figure 7.** Location of the study sites chosen for the validation of the MLake model; (a) Rhône, (b) White Nile, (c) Angara, and (d) Neva. Made with Natural Earth topographic maps.

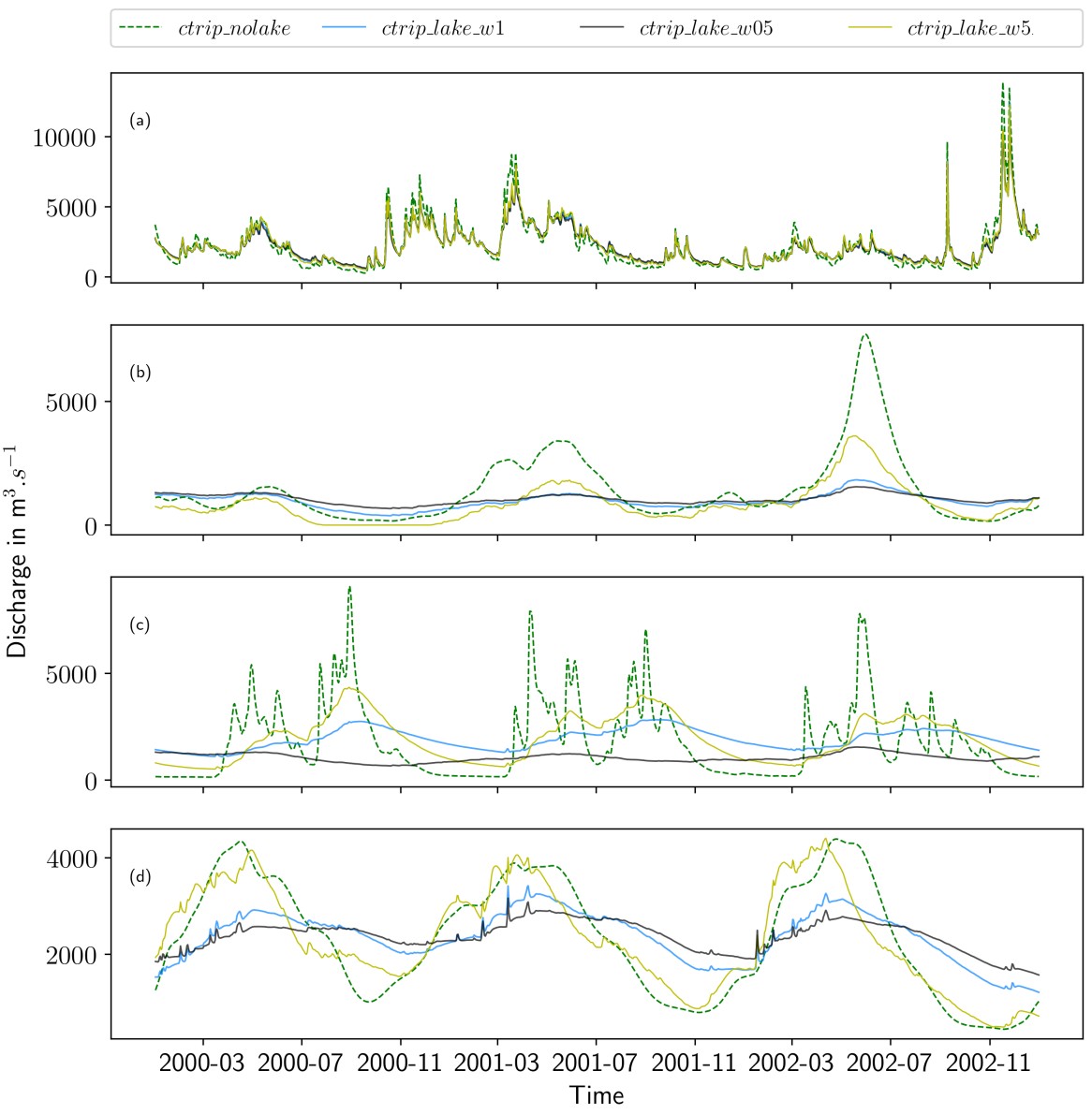

**Figure 8.** Hydrograph of the simulated river discharge over the period 2000-2002 for the different CTRIP-MLake configurations; a) Rhône, b) Angara, c) White Nile, and d)Neva.

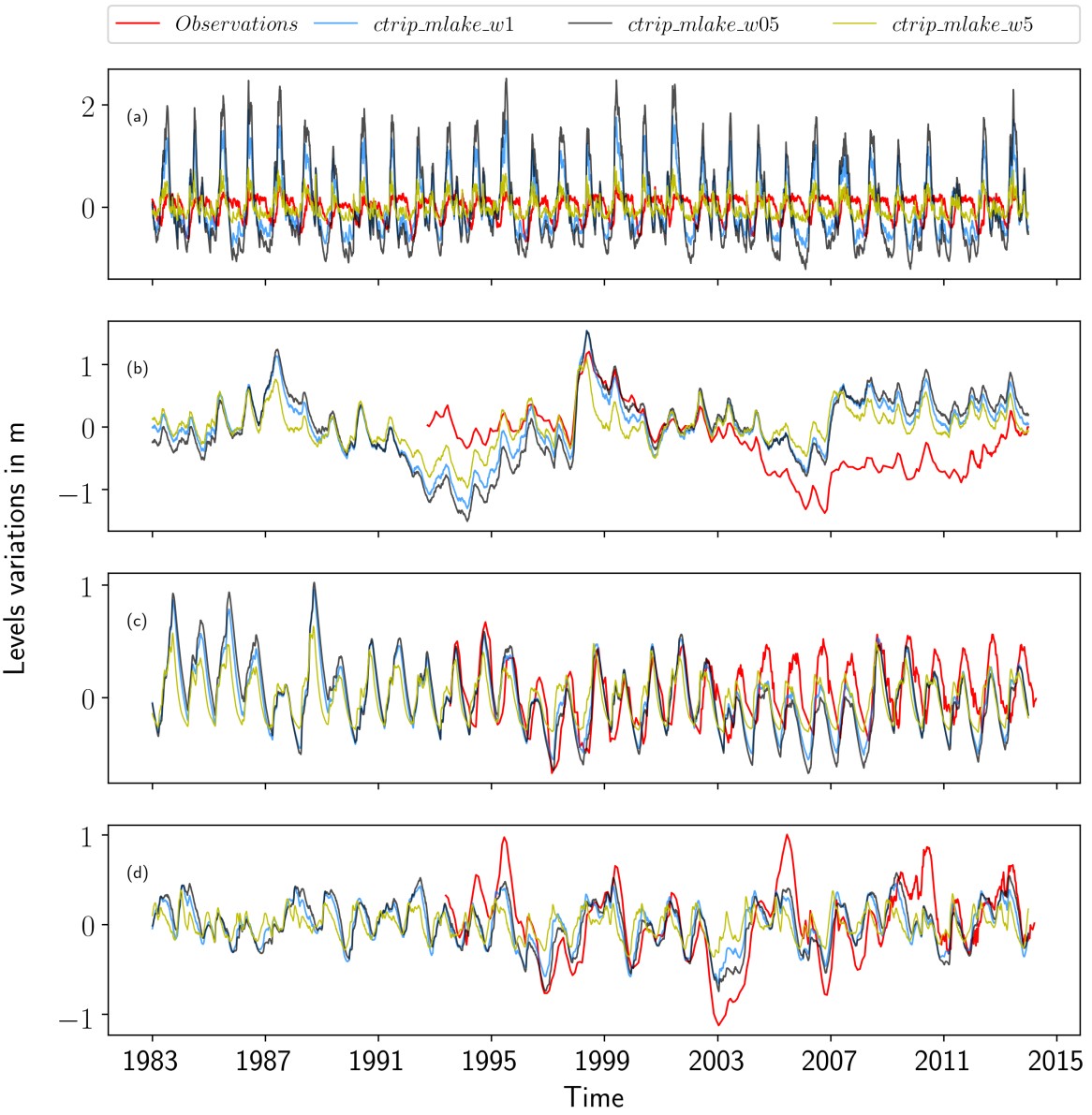

**Figure 9.** Simulated and observed lake level variations over the period 1983-2014 in the different CTRIP-MLake configurations. (a) Lake Geneva, (b) Lake Victoria (c) Lake Baikal (d) Lake Ladoga

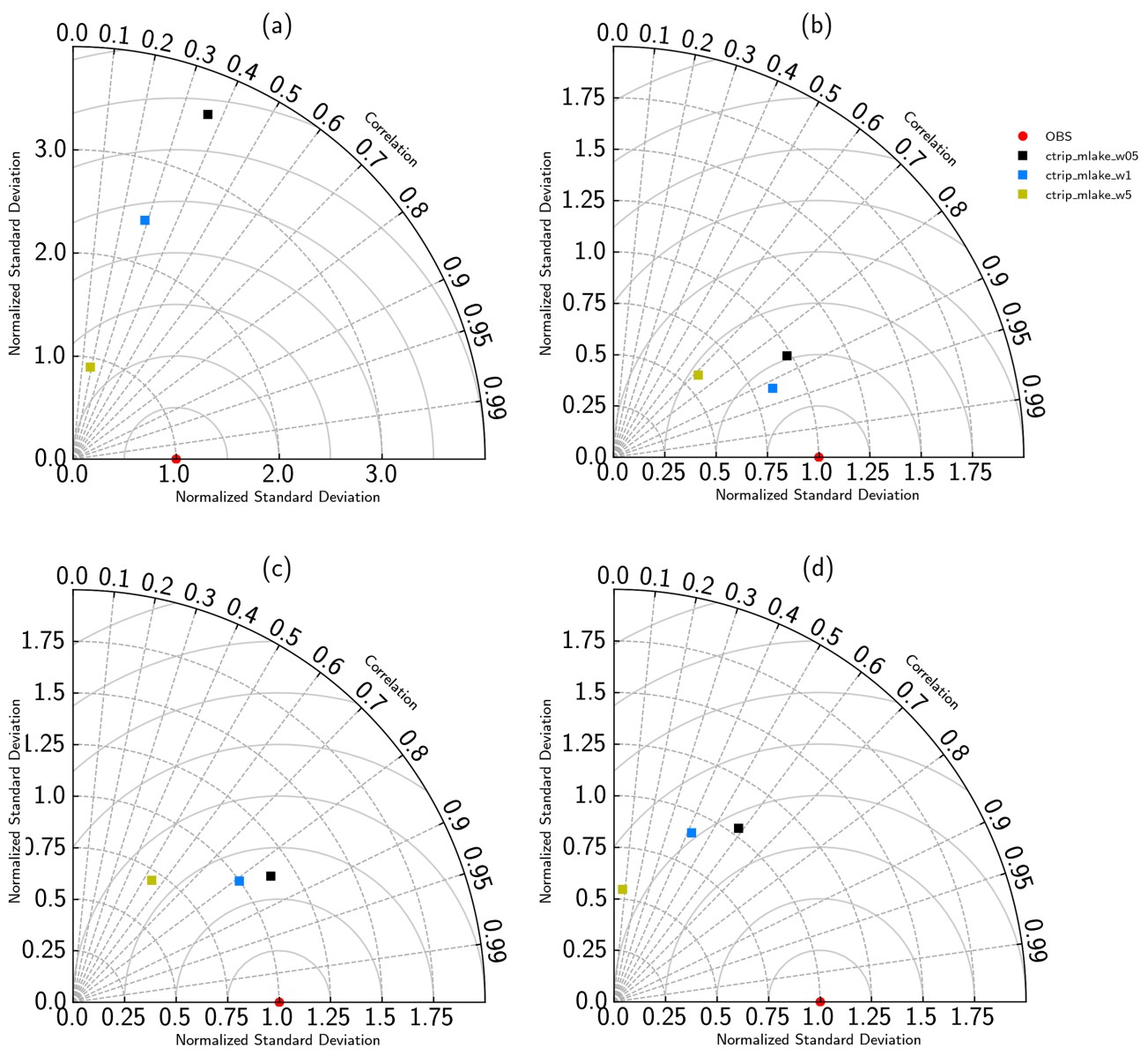

**Figure 10.** Taylor diagram showing the simulated and observed lake levels variations scores over the period 1983-2004 for Lake Victoria and 1983-2014 for the three others. (a) Lake Geneva, (b) Lake Victoria (c) Lake Baikal (d) Lake Ladoga

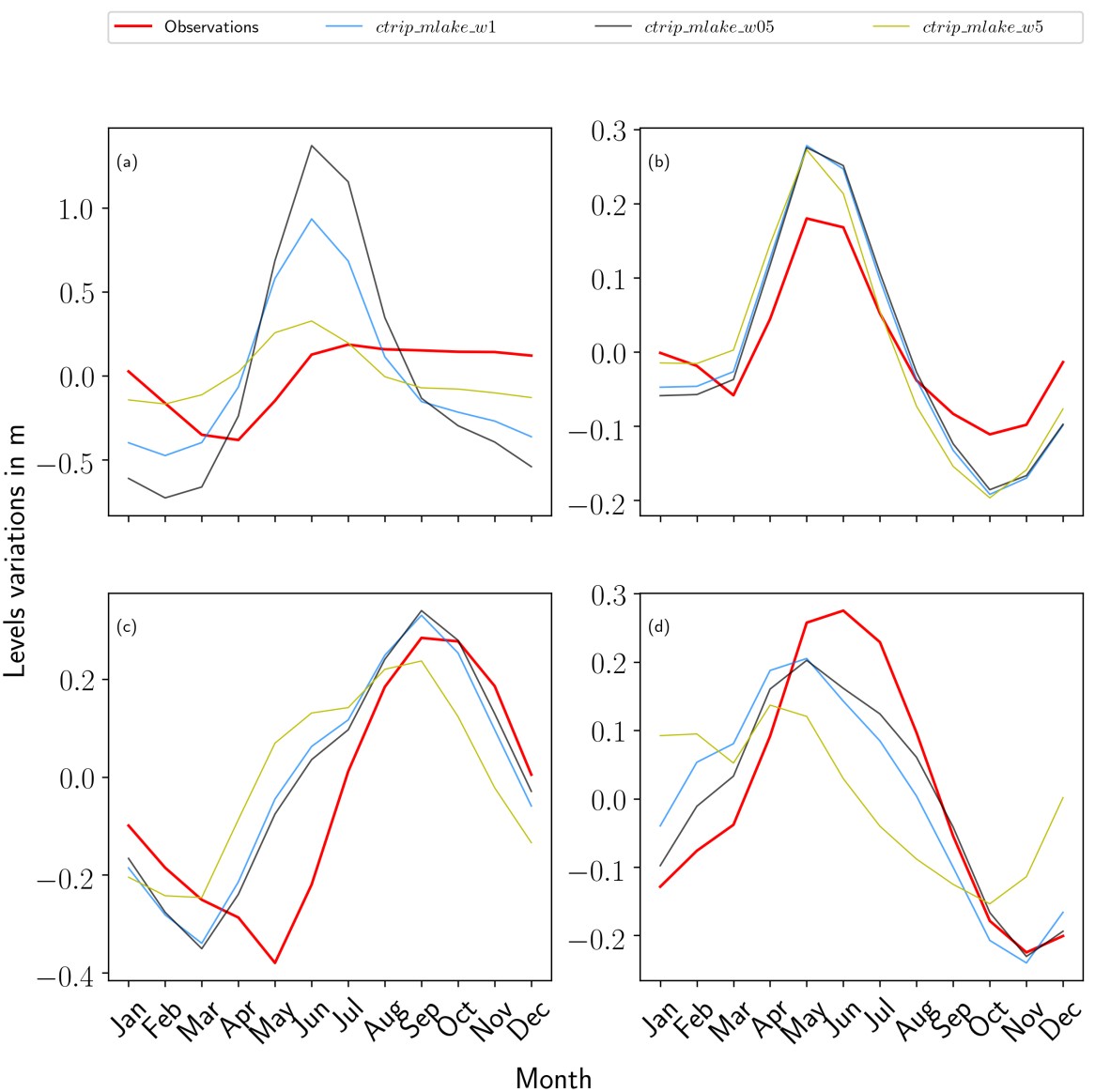

**Figure 11.** Seasonal mean of lake level variations over the period 1983-2004 for Lake Victoria (b) and for 1983-2014 for the three others; (a) Lake Geneva, (c) Lake Baikal, and (d) Lake Ladoga

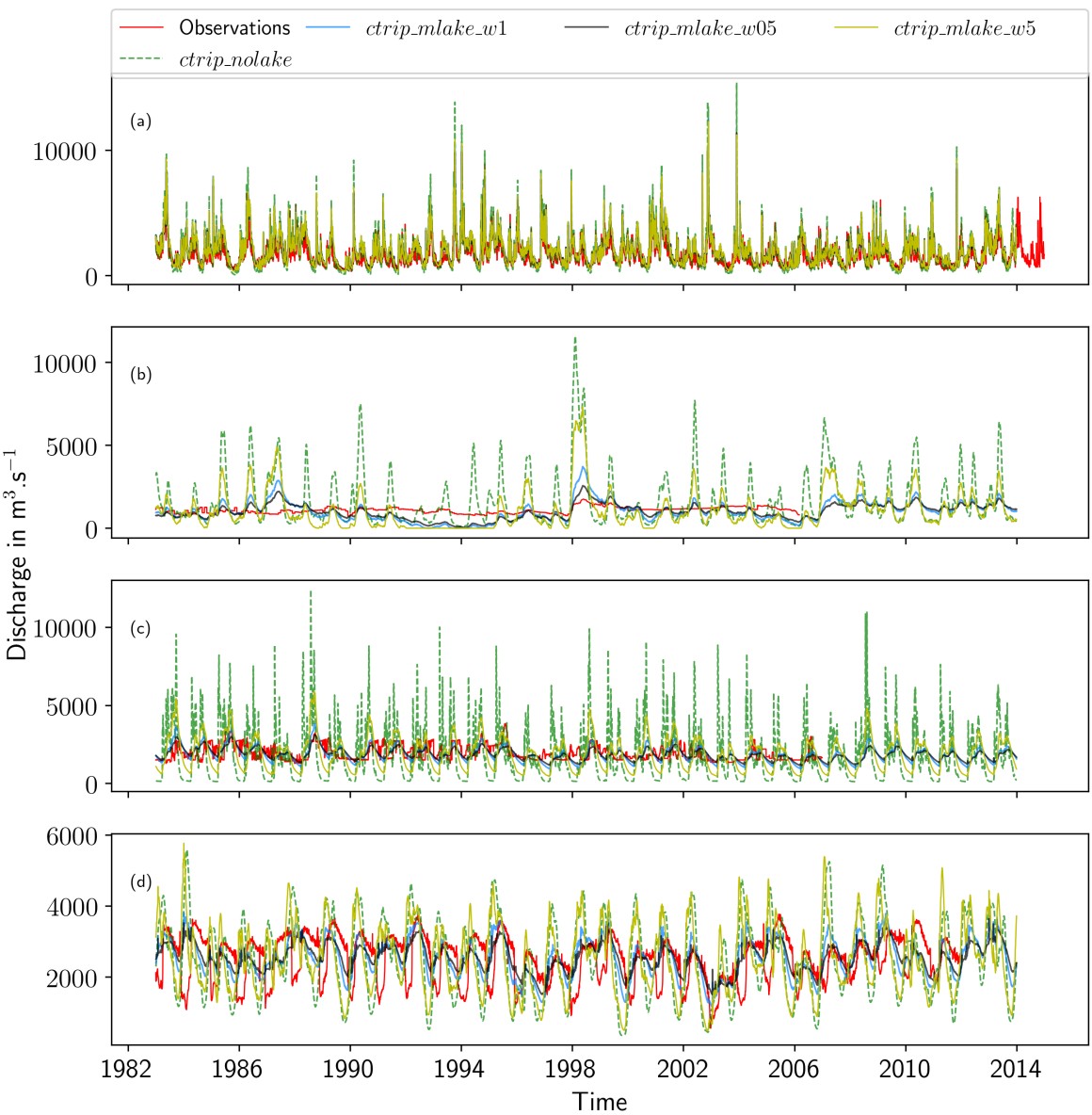

**Figure 12.** Hydrograph of the different rivers at their control station over the period 1983-2014 in the different CTRIP-MLake configurations; (a) Lake Geneva, (b) Lake Victoria, (c) Lake Baikal, and (d) Lake Ladoga.

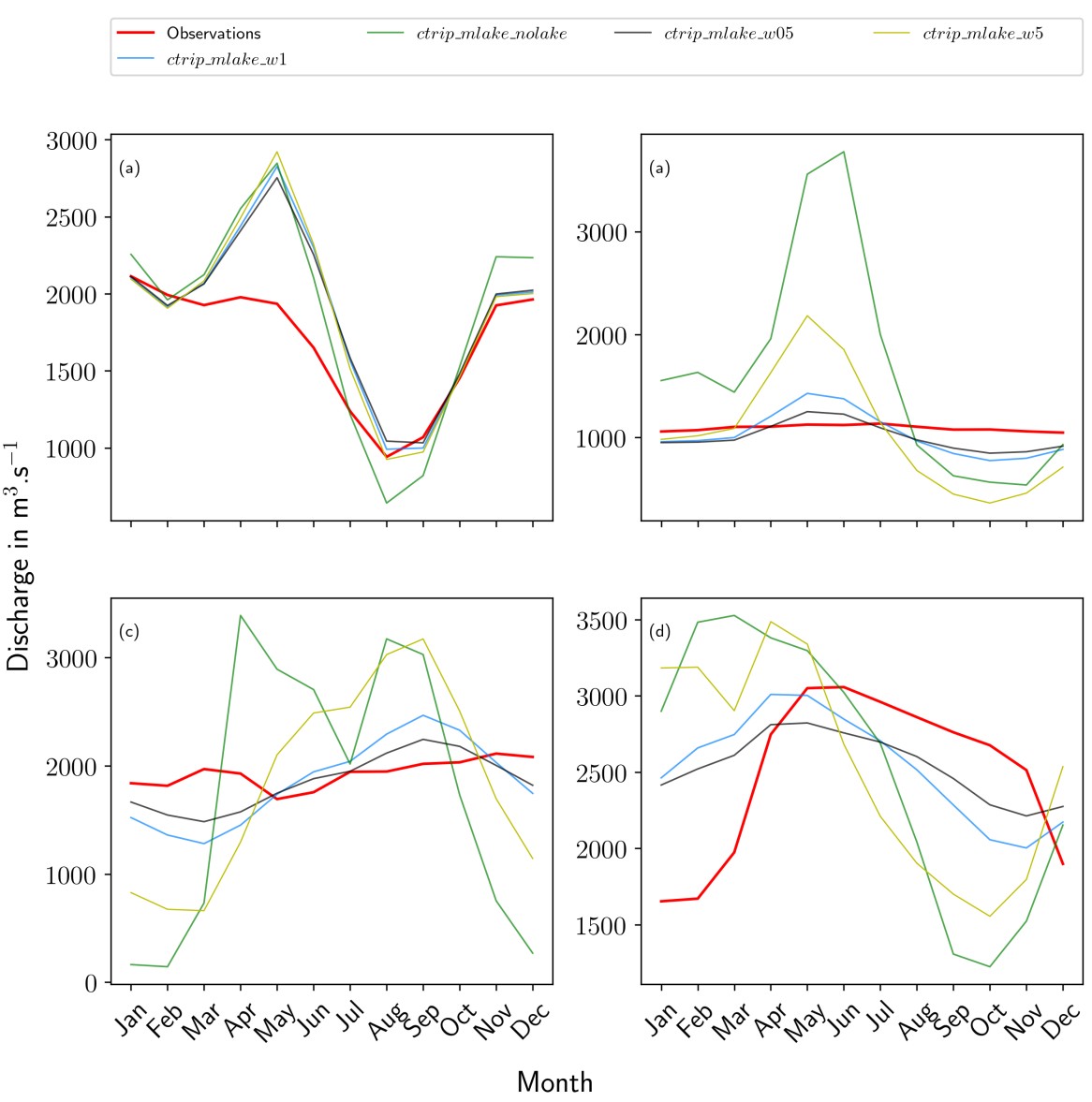

**Figure 13.** Seasonal mean of the observed and simulated CTRIP-MLake discharge over the period 1983-2013 for (a) Lake Geneva, (b) Lake Victoria, (c) Lake Baikal, and (d) Lake Ladoga

**Table 1.** Land surface model integrating a mass balance lake parametrization

| Model Name | Reference |
|---|---|
| Community Land Model | Oleson et al. (2010) |
| Variable Infiltration Capacity | Bowling and Lettenmaier (2010); Mishra et al. (2010) |
| LISFLOOD | Burek et al. (2013); Zajac et al. (2017) |
| MESH | Pietroniro et al. (2007) |

**Table 2.** Lake parameters and variables introduced in CTRIP scripts

|  |  |  |  |
|---|---|---|---|
| Parameters | $lake\_id$ | - | lake id number on the runoff mask |
|  | $lake\_net\_id$ | - | lake id number on the network mask |
|  | $frac\_lake$ | - | lake fraction on every CTRIP 12D pixel grid |
|  | $z\_mean$ | m | lake mean depth from GLDB database |
|  | $a\_lake$ | m$^2$ | lake surface area from ECOCLIMAP-II |
|  | $weir\_z$ | m | crest height at the lake outlet |
|  | $weir\_w$ | m | lake outlet width |
| Variables | $lake\_sto$ | $kg$ | lakes storage |
|  | $lake\_h$ | m | lake level height over the mean initial depth |
|  | $lake\_out$ | $kg.s^{-1}$ | lake outflows over the weir |

**Table 3.** Description of the study site chosen for the evaluation of MLake

| Watershed 1: Rhône | |
|---|---|
| Basin name | Rhône |
| Outlet chosen | Beaucaire (France) |
| Drainage area | 97,800 $km^2$ |
| Number of lakes considered | 5 |
| Main lake | Lake Geneva |
| Köppen-Gegeir climate classification for the main lake | Cfb (Temperate continental climate with cool humid winter and relative cool summer) |
| **Watershed 2: Angara** | |
| Basin name | Angara |
| Outlet chosen | Irkoutsk (Russia) |
| Drainage area | 577,000 $km^2$ |
| Number of lakes considered | 9 |
| Main lake | Lake Baikal |
| Köppen-Gegeir climate classification for the main lake | Dwb (Cold continental climate with dry winter and temperate summer) |
| **Watershed 3: White Nile** | |
| Basin name | White Nile |
| Outlet chosen | Jinja (Uganda) |
| Drainage area | 167,000 $km^2$ |
| Number of lakes considered | 1 |
| Main lake | Lake Victoria |
| Köppen-Gegeir climate classification for the main lake | Af (Equatorial climate) |
| **Watershed 4: Neva river** | |
| Basin name | Neva |
| Outlet chosen | Saint-Petersburg (Russia) |
| Drainage area | 97,800 $km^2$ |
| Number of lakes considered | 5 |
| Main lake | Lake Ladoga |
| Köppen-Gegeir climate classification for the main lake | Dfb (Warm summer continental climate) |

**Table 4.** Configuration of the different runs chosen for the study

| Name of the study site | Simulation | Forcing data | Details |
|---|---|---|---|
| Rhône | $ctrip\_nolake$ | SAFRAN-ISBA | Reference ISBA-CTRIP simulation without the lake model |
| | $ctrip\_mlake\_w1$ | SAFRAN-ISBA | ISBA-CTRIP-MLake simulation with an initial $weir\_w$ equal to the downstream river width |
| | $ctrip\_mlake\_w0.5$ | SAFRAN-ISBA | ISBA-CTRIP-MLake simulation with the initial $weir\_w$ divided by two |
| | $ctrip\_mlake\_w5$ | SAFRAN-ISBA | ISBA-CTRIP-MLake simulation with the initial $weir\_w$ multiplied by five |
| White Nile, Angara, Neva | $ctrip\_nolake$ | Earth2Observe | Reference ISBA-CTRIP simulation without the lake model |
| | $ctrip\_mlake\_w1$ | Earth2Observe | ISBA-CTRIP-MLake simulation with an initial $weir\_w$ equal to the downstream river width |
| | $ctrip\_mlake\_w0.5$ | Earth2Observe | ISBA-CTRIP-MLake simulation with the initial $weir\_w$ divided by two |
| | $ctrip\_mlake\_w5$ | Earth2Observe | ISBA-CTRIP-MLake simulation with the initial $weir\_w$ multiplied by five |

**Table 5.** Results on the daily river discharge

| River | Run name | $\bar{Q}$ $(m^3.s^{-1})$ | $\sigma$ $(m^3.s^{-1})$ | Relative distance $\sigma$ to CTRIP reference simulation |
|---|---|---|---|---|
| Rhône | ctrip_nolake | 1876 | 1371 | - |
| | ctrip_mlake_w1 | 1889 | 1064 | -0.22 |
| | ctrip_mlake_w0.5 | 1889 | 1036 | -0.24 |
| | ctrip_mlake_w5 | 1889 | 1106 | -0.19 |
| | observations | 1703 | 1003 | - |
| White Nile | ctrip_nolake | 1625 | 1631 | - |
| | ctrip_mlake_w1 | 1031 | 613 | -0.62 |
| | ctrip_mlake_w0.5 | 1005 | 455 | -0.72 |
| | ctrip_mlake_w5 | 1047 | 1139 | -0.30 |
| | observations | 1057 | 300 | - |
| Angara | ctrip_nolake | 1754 | 1696 | - |
| | ctrip_mlake_w1 | 1852 | 498 | -0.70 |
| | ctrip_mlake_w0.5 | 1852 | 356 | -0.79 |
| | ctrip_mlake_w5 | 1850 | 1021 | -0.40 |
| | observations | 1860 | 486 | - |
| Neva | ctrip_nolake | 2541 | 1095 | - |
| | ctrip_mlake_w1 | 2538 | 544 | -0.50 |
| | ctrip_mlake_w0.5 | 2539 | 394 | -0.64 |
| | ctrip_mlake_w5 | 2537 | 964 | -0.11 |
| | observations | 2485 | 655 | - |

**Table 6.** Performance metrics comparison for the daily simulated and observed river discharges for the study sites.

| River | Run name | NSE | NSE log | Correlation | NIC | $\overline{Q_s/Q_o}$ | $\sigma_s/\sigma_o$ |
|---|---|---|---|---|---|---|---|
| Rhône | *ctrip_nolake* | 0.58 | 0.34 | 0.90 | - | 1.10 | 1.37 |
| | *ctrip_mlake_w1* | 0.69 | 0.64 | 0.87 | 0.26 | 1.11 | 1.06 |
| | *ctrip_mlake_w0.5* | 0.71 | 0.66 | 0.88 | 0.31 | 1.11 | 1.03 |
| | *ctrip_mlake_w5* | 0.66 | 0.62 | 0.86 | 0.19 | 1.11 | 1.1 |
| White Nile | *ctrip_nolake* | -84.4 | -30.9 | 0.27 | - | 1.45 | 5.6 |
| | *ctrip_mlake_w1* | -9.6 | -4.6 | 0.47 | 0.88 | 0.87 | 2.1 |
| | *ctrip_mlake_w0.5* | -4.7 | -3.8 | 0.53 | 0.93 | 0.86 | 1.54 |
| | *ctrip_mlake_w5* | -38.4 | -16.5 | 0.33 | 0.54 | 0.87 | 3.9 |
| Angara | *ctrip_nolake* | -12.2 | -36.1 | 0.08 | - | 0.94 | 3.49 |
| | *ctrip_mlake_w1* | -0.1 | -0.33 | 0.48 | 0.92 | 0.995 | 1.02 |
| | *ctrip_mlake_w0.5* | 0.20 | 0.12 | 0.51 | 0.94 | 0.995 | 0.92 |
| | *ctrip_mlake_w5* | -3.4 | -6. | 0.28 | 0.75 | 0.99 | 2.1 |
| Neva | *ctrip_nolake* | -2.94 | -3.2 | -0.02 | - | 1.02 | 1.67 |
| | *ctrip_mlake_w1* | -0.08 | -0.03 | 0.37 | 0.73 | 1.02 | 0.83 |
| | *ctrip_mlake_w0.5* | 0.26 | 0.23 | 0.52 | 0.81 | 1.02 | 0.60 |
| | *ctrip_mlake_w5* | -2.3 | -2.2 | -0.04 | 0.15 | 1.02 | 1.47 |

**Table 7.** Performance metrics comparison of the simulated and observed lake level variations on the study sites

| Lake | Run name | $R^2$ | $\sigma_o$ | $\sigma_s(CV)$ | RMSE (m) |
|---|---|---|---|---|---|
| | $ctrip\_mlake\_w1$ | 0.29 | | 0.54 (2.45) | 0.52 |
| Lake Geneva | $ctrip\_mlake\_w0.5$ | 0.37 | 0.22 | 0.80 (3.6) | 0.75 |
| | $ctrip\_mlake\_w5$ | 0.19 | | 0.20 (0.9) | 0.27 |
| | $ctrip\_mlake\_w1$ | 0.92 | | 0.31 (0.54) | 0.31 |
| Lake Victoria | $ctrip\_mlake\_w0.5$ | 0.86 | 0.57 | 0.33 (0.58) | 0.33 |
| | $ctrip\_mlake\_w5$ | 0.72 | | 0.33 (0.58) | 0.43 |
| | $ctrip\_mlake\_w1$ | 0.82 | | 0.27 (0.96) | 0.23 |
| Lake Baikal | $ctrip\_mlake\_w0.5$ | 0.86 | 0.28 | 0.31 (1.1) | 0.26 |
| | $ctrip\_mlake\_w5$ | 0.59 | | 0.19 (0.68) | 0.27 |
| | $ctrip\_mlake\_w1$ | 0.42 | | 0.24 (0.92) | 0.28 |
| Lake Ladoga | $ctrip\_mlake\_w0.5$ | 0.58 | 0.26 | 0.27 (1.03) | 0.24 |
| | $ctrip\_mlake\_w5$ | 0.07 | | 0.14 (0.54) | 0.35 |

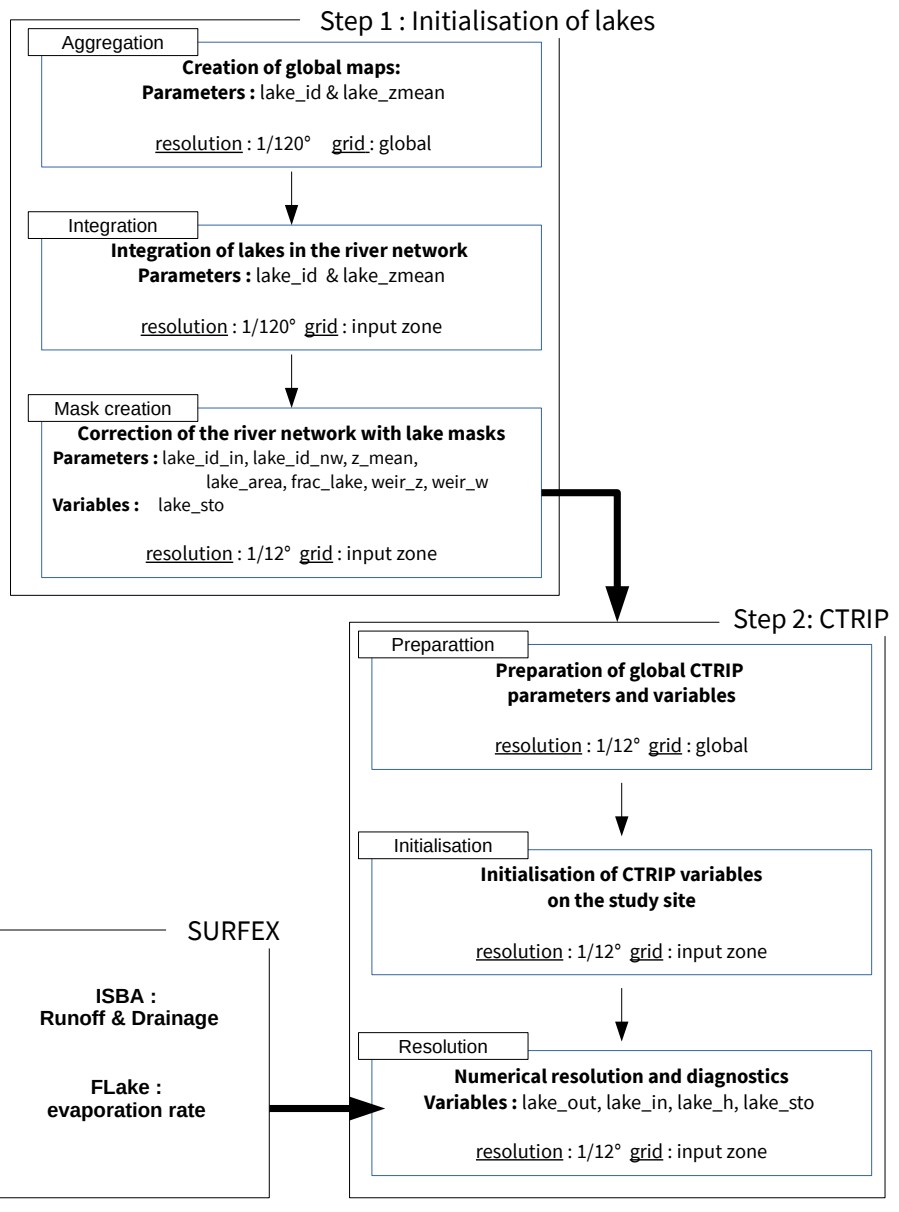

**Figure A1.** Road map of the ISBA-CTRIP-MLake modelling system. In each step, the name of the routine and their extension are in the upper left square, bold text represents the binary or netcdf output files used.