# Peer review of "Parametrization of a lakes water dynamics model MLake in the ISBA-CTRIP land surface system (SURFEX v8.1)"

_Geoscientific Model Development, 2020_

## Referee Comment (RC1) · Dai Yamazaki (Referee) · 8 Dec 2020

<General Comments>

This manuscript presents the new lake mass balance scheme (MLAKE) in ISBA-CTRIP model. As lake is usually not explicitly treated in the land surface component of global climate models, this work on the description of lake model development and analysis has a potential contribution to Earth system modelling community. I found the model description is well written, while some improvements are needed mainly on the presentation quality before publication. Detailed comments are summarized below.

<Specific Comments>

Abstract: First paragraph

This paragraph on the research background is too long as a part of the abstract. Please think about shortening this by summarizing an important point.

L44: Lakes are of fundamental importance to ensure

The first paragraph is too long. You can start the second paragraph from this sentence, as importance of lakes is explained from this part, while the previous part discusses about hydrological cycle in climate system.

L50: Where present, lakes play a triple role in the Earth system. . .

Again, you can start a new paragraph here, focusing on the roles of the lake. Probably, you can explain each of the three roles in a separated paragraph, as the explanation of each role contains several sentences. This will largely increase readability. Also, it is very difficult to guess what is "triple roles" only from this sentence. This sentence only contains: 1) energy and water balance in GCM, 2) impact on local climate and hydrology. Based on the following discussion, "3) interaction with biogeochemical cycle" is missing from this sentence.

L52: First, they influence...

In addition to the lake surface impact on atmosphere, the changes in timing and volume of freshwater discharge to oceans might affect both local and global ocean circulations. This is better to be included as the first role of lakes in Earth System.

L57 "Economic lever"

Is it appropriate to use the word "economic" in this context? We may say "ecological lever". If "economic lever" is more suitable, please explain.

L98: In the recent years, many studies have focused on

Again, I think this paragraph is too long, and you can start a new sentence from here focusing on recent studies on lake representation.

L139: Section 5 shows the

Description of "Section 4" is missing.

L189: Manning equation

Strictly, Manning's equation is to give friction energy loss. Flow velocity is not directly given only from the Manning's roughness equation, and it should be "kinematic wave approximation of shallow water equation, with Manning's roughness equation as friction energy term).

L205: This resolution framework assures the resolution is done

It is not clear what the authors want to mean by the frase "the resolution is done". Please explain.

L274: The approach used herein to resolve this issue is

How this modification was done? Is this done manually? If so, how long did it take correct the issues in the test domains of this study? Is it feasible to perform similar amount of modifications at a global scale? Please discuss.

L425: which is the second largest lake

You need to add "in terms of surface area", because the size of lakes are explained in different metrics throughout the manuscript.

L465: Atmospheric forcing

Please explain why two different forcing datasets are used in this study? What is the purpose of using different forcing only for France?

L483: Biases may appear in simulated surface/sub-surface variables

This sentence is confusing, as this is "excuse" to use multiple forcing datasets. It is better to state that "multiple forcing datasets are used in this study" for better understanding of this paragraph.

L513: with an average discharge increased by 0.7 %

Is it reasonable that the average discharge increased? What was the background physics mechanism? Please explain. I guess this could be due to the change in discharge timing, and it changes the discharge at the start and/or end of the simulation period resulting in a slight difference in the total discharge. In this case, the 0.7% increase is negligible and better to state "average discharge is not affected" rather than "0.7% increased".

L525: where the weir width is increased by a factor of five compared to ctrip_mlake_w1.

I wonder whether this simulation setting (500% of the control experiment) is reasonable or not. Given that the "lake outlet width" is observable parameter, there must be a reasonable range for this parameter. I think the sensitivity test should be designed within this "reasonable parameter range".

L620: lake level variations ($sigma\_s = 1069$ m3s$-1$, $sigma\_o = 1003$ m3s$-1$).

Is the unit (m3/s) correct? As this sentence mentioned for lake level.

L633: The NIC score has. . .

What is "NCC score"? It once appears in the abstract first, and appears first time here in the maintext without any definition or explanation. Please provide the description of NIC score.

L666: the worlds largest freshwater continental water body

Please explain "largest in terms of what?"

L693: with a net decrease in the peak discharge.

What does "net decrease in the peak discharge"? Usually the term "net" is used for the total summation, but the term is used for "peak (maximum)" here. How can we define "net decrease in the peak"?

L717: Simulations reveal the capability of the non-calibrated CTRIP-MLake

This paragraph mainly discusses the impact of lake internal dynamics caused by wind, rather than discussing the overall limitation of the proposed model. The lake internal dynamic parts is better to be shown as a separate sub-section (i.e. 6.1 lake internal dynamics), as the following discussion points are explicitly shows with sub section title. I also suggest that the most significant difficulty of the internal height variation appears in the comparison between modeled and observed water levels. This point should be discussed explicitly.

In addition, freezing of lake surface could cause significant difference in simulations. Is this represented in the current model? If not, better to explain as one of the major limitations.

L764: Historical Lake Victoria level drops

I think it is better to remove this "sub-section title", to keep the consistency with other paragraphs.

L865: the lake outlet size

The term "lake outlet size" only appears here. Please use "lake outlet width" to keep consistency.

L871: Finally.

It is a bit strange to see "Finally" after the sentence starting from "Last but not least".

Figure 3:

Please change the figure caption from French to English.

Figure 4:

It is difficult to know what the authors want to discuss with this figure. Please explain what does this figure want to explain in the caption, for better understanding of the

river/lake map preparation.

Figure 9:

Is it possible to add observed discharge in this figure?

———————————————————

---

## Referee Comment (RC2) · Anonymous Referee #2 · 8 Dec 2020

**Review for Parameterization of lakes water dynamics in the ISBA-CTRIP land surface system by Guinaldo et al.**

**Paper summary:**

The paper describes new lake mass balance model component called MLake and how the new module is integrated into the river routing model (CTRIP) coupled to ISBA land surface model. The paper also evaluated the developed model at 4 river basins across the globes and shows improvement of downstream discharges and lake level simulations compared to the observations.

**Overall comments:**

The descriptions of lake water balance model implementation into a river model for ESM fits very well in the GMD, model description paper. The lake water balance computation used in this paper seems to be more precise (account for lake-ground water interaction, separate treatment for surface runoff inflow and baseflow inflow) than other lake models even used for global hydrologic models. So, particularly for the ESM, I think it is substantial advancement of model process representations.

Simulated impacts of lake parameter (i.e., outlet structure parameter) on the lake level and downstream discharge seems to be reasonable and discussed well. Figures showing results are also clear overall.

My main concern is a lack of conciseness: the paper can be shortened by textural editing as well as by cutting some materials. More importantly, I feel the authors need to polish the descriptions throughout the paper. I put a few science related comments following numerous minor editorial comments I found as I read.

**Specific comments:**

- P29-L920: This might be typo, but KGE expression seems to be incorrect. Standard KGE (Gupta et al., 2009) is a distance based on correlation (r), ratio of standard deviation (alpha) and mean ratio (beta). Modified KGE (from Kling et al. 2012) uses ratio of coefficient of variation (gamma) instead of ratio of standard deviation (alpha).

  In my opinion, KGE is convenient for model calibration since KGE aggregates three metrics into one, which allow modelers to use a single objective (target metric) to optimize the model, rather than multi-objectives, but for just model evaluation, KGE (aggregated metric) itself does not mean much (what aspects of time series the model simulate better or worse). Since table 6 compare Qs/Qo and sigma_s/sigma_o, I would suggest adding correlation and removing KGE.

- Section 2.3.2 and 2.3.3 seems be a crux of the paper (descriptions of lake implementation strategy), and therefore need to be clear so reader can understand how exactly the lake are implemented in the network. Unfortunately, I am having hard time following section 2.3.2 after I read through several times. So, I feel I need to request the author revise this section drastically. Section 2.3.3 is described well.

- P22, L717: I found this (sub-grid variability of lake levels for a large lake) is interesting discussion and may be important. Impact of such lake sub-grid variability on downstream discharge is described in L725, but not clear and would be nice to see a little more elaboration on this. For even larger lakes

(e.g., Great Lakes), the lakes should be represented by a number of grid boxes, and make individual grided lake interact each other?

**Editorial comments:**

- Throughout the paper:  The authors use "resolution", instead of "solution", which I would suggest using (e.g., numerical solution, not numerical resolution). Using "resolution" to mean "solve an equation or something" be confusing since "resolution" is also used for grid box sizes and vertical resolution of soil layer, snow layers.

- Throughout the paper: I know this may not be critical issue…but I feel the paper uses "scale" carelessly. "scale" is the characteristic length (space and time) of processes, modeling and observations (see Blöschl et al., 1995: https://onlinelibrary.wiley.com/doi/abs/10.1002/hyp.3360090305) . For example, "global scale model" used throughout the paper means globe is the characteristic length of modeling, but it really means global domain model and scale used for model should be smaller.

- Title:  SURFEX v8.1 is the name of land surface model platform, but here, should lake model component name and version be in title. Here, is MLake name?

- P2, L44-46: I feel this sentence is really opening sentence. Descriptions priors to this sentence seem to be little related to the topic in this paper.

- P3, L65: Suggest adding references. Climate change impact on Lake Chad has been reported in publication (https://www.nature.com/articles/s41598-020-62417-w)

- P3, L70: Suggest changing runoff -> discharge. e.g., lowering inter-annual and seasonal variability of downstream discharge?

- P3, L95: "General Hydrological Model" -> Global Hydrological Model?? If not, what does General Hydrologic model mean?

- P5, L136: Awkward description. Suggest removing this.

- P7, L214: main driver -> main motivation?

- P8, L258: what does "component" mean here?  I believe it is lake, but please be specific.

- P8, L272: "dynamic close to a lake".  Not clear to me.

- P8, L273: "lake hydrological dynamic".  Sounds awkward to me.

- section 2.2: this section provides brief description of Flake model, which simulates energy balance in lake (my understanding), and lake evaporation is a part of the energy balance. I feel header should be "Flake model: lake energy balance model". This way, consistent with the following section (MLake)

- Section 2.3.4: I am not sure about importance of this section. Figure 7 could follow more convention of flowchart (https://en.wikipedia.org/wiki/Flowchart).  I wonder if this section (after shortened) could be moved to very beginning of section 2.

- P22, L719: "uni-dimensional" -> one-dimensional.

- P22, L724: "fast time variations of the river discharge". Not clear phrase. Please consider describing different way.

- P25, L798: "a unique composite energy budget for soil and vegetation". Does this mean control volume for energy budget computation is combined of soil and vegetation (not separate)?

- P26, L834-836: this paragraph sounds out of place in this section (simulation sensitivity to lake outlet width)

- P27, L842: "has been conducted" -> use past tense?

- P27, L842: "four river networks" -> four river basins?

- P27, L865: "monitoring" -> monitor.

- P27, L867-869: I would suggest moving this sentence to the end of the paragraph (replace the last sentence with this). Eventually we would like to see the effect of lake bathymetries on lake levels, downstream discharge across the globes.

- P28, L878-L879: I would suggest adding references. Some groups have done some work on reservoir operation schemes, e.g., Hanasaki et al., 2006 JH, Shin et al., 2020 WRR.

- P28: Appendix header is missing?

- Figure 11. Add y-axis labels.

-

---

## Author Comment (AC1) · 28 Jan 2021

All of the authors and co-authors would like to thank the referee for the time which he/she has allocated to the detailed revision of this paper and her/his positive comments about our work that we feel have helped us to make an improved version of the manuscript. Responses to comments and subsequent changes are detailed below in blue.

Please also note the supplement to this comment:
https://gmd.copernicus.org/preprints/gmd-2020-296/gmd-2020-296-AC1-supplement.pdf

[Figure]

**Supplement:**

\<General Comments\>

This manuscript presents the new lake mass balance scheme (MLAKE) in ISBA-CTRIP model. As lake is usually not explicitly treated in the land surface component of global climate models, this work on the description of lake model development and analysis has a potential contribution to Earth system modelling community. I found the model description is well written, while some improvements are needed mainly on the presentation quality before publication. Detailed comments are summarized below.

All of the authors and co-authors would like to thank the referee for the time which he/she has allocated to the detailed revision of this paper and her/his positive comments about our work that we feel have helped us to make an improved version of the manuscript. Responses to comments and subsequent changes are detailed below in blue.

\<Specific Comments\>

Abstract: First paragraph

This paragraph on the research background is too long as a part of the abstract. Please think about shortening this by summarizing an important point.

L3-5 have been deleted in order to shorten the abstract has been shortened.

L44: Lakes are of fundamental importance to ensure
The first paragraph is too long. You can start the second paragraph from this sentence, as importance of lakes is explained from this part, while the previous part discusses about hydrological cycle in climate system.

We have removed the first paragraph and modified the introduction sentence with :

P.2, L.28
« Only 2.5 % of the total water mass of the planet is defined as fresh water, and only a very small fraction is directly accessible for human consumption (Oki and Kanae, 2006). Lakes are of fundamental importance to ensure freshwater supply to the 800 million people which have insufficient safe drinking water, according to the World Health Organization (WHO, 2010; Marsily et al., 2018).»

L50: Where present, lakes play a triple role in the Earth system…

Again, you can start a new paragraph here, focusing on the roles of the lake. Probably, you can explain each of the three roles in a separated paragraph, as the explanation of each role contains several sentences. This will largely increase readability. Also, itis very difficult to guess what is "triple roles" only from this sentence. This sentence only contains: 1) energy and water balance in GCM, 2) impact on local climate and hydrology. Based on the following discussion, "3) interaction with biogeochemical cycle"is missing from this sentence.

Corrections related to the organisation have been added (addition of new paragraphs) and we have made an effort to improve the readability of the three roles.
These three roles are 1) change in surface energy fluxes and ocean circulation, 2) impact the climate from local to continental scale, 3) interaction with the regional water cycle

L52: First, they influence…

In addition to the lake surface impact on atmosphere, the changes in timing and volume of freshwater discharge to oceans might affect both local and global ocean circulations. This is better to be included as the first role of lakes in Earth System.

In response, we have added the potential impact of lake fluxes on local and global ocean circulations.
P.2, L39;

In addition, lakes influence the freshwater flux variability which in the end interact with the local (Sauvage et al, 2018) and global ocean circulation (Rahmstorf, 1995).

L57 "Economic lever"

Is it appropriate to use the word "economic" in this context? We may say "ecological lever". If "economic lever" is more suitable, please explain.

The lake ecological importance has been added in the paragraph.

« Second, as sentinels of climate change, lakes must not only be seen as water reservoirs but also as a major ecological levers. »

However, lakes are important sources of socio-economic development in the related regions through the ecosystem services that population take back from these water bodies.
We added a note P.3, L73 and a reference to this : Schallenberg et al, 2013. Ecosystem services of lakes.

L98: In the recent years, many studies have focused on

Again, I think this paragraph is too long, and you can start a new sentence from here focusing on recent studies on lake representation.

P3. L89
We started a new paragraph to improve the flow of the introduction

L139: Section 5 shows the
Description of "Section 4" is missing.

As suggested by Referee#2, this section has been removed.

L189: Manning equation

Strictly, Manning's equation is to give friction energy loss. Flow velocity is not directly given only from the Manning's roughness equation, and it should be "kinematic wave approximation of shallow water equation, with Manning's roughness equation as friction energy term).

We totally agree on the lack of clarity in this sentence . Changes have been made:
P.6, L.173
« Streamflow routing is simulated using CTRIP (Fig.1) which integrates a dynamic computation of river flows based on a kinematic
wave approximation which is solved using  Manning's roughness equation as a friction energy dissipation term which is dependent on the characteristics of the river section. »

L205: This resolution framework assures the resolution is done

It is not clear what the authors want to mean by the frase "the resolution is done".Please explain.

This sentence is unclear due to the mismatch in the definition of resolution. The aim was to provide information on the capability of the numerical computation to ensure that all of the upstream portions of the network have been resolved before solving the equations in a new node. In order to improve the description, we changed the sentence with the following :

P.6, L191
« This numerical solution framework assures the computation of river discharge is performed starting from the upstream cells and then progressing to the downstream cells of the watershed. In every basin, the head-water cells have the lowest sequence order: one, which is incremented for each downstream cell. »

L274: The approach used herein to resolve this issue is

How this modification was done? Is this done manually? If so, how long did it take correct the issues in the test domains of this study? Is it feasible to perform similar amount of modifications at a global scale? Please discuss.

Thank you for raising this interesting point. In our approach, the modification is done numerically within the code by adding the considered threshold. The modification is thus applicable to both local and global scales.

P.9, L261
« The approach used herein to resolve this issue is to replace  a river section by a lake pixel (corresponding to a unique node in the network) when a lake covers at least 50 % of a given grid cell »

L425: which is the second largest lake
You need to add "in terms of surface area", because the size of lakes are explained indifferent metrics throughout the manuscript.

Correction has been made by adding the proposed wording.

L465: Atmospheric forcing

Please explain why two different forcing datasets are used in this study? What is the purpose of using different forcing only for France?

Thank you for this valuable comment. The sentence has been changed in order to better explain the reason for choosing different datasets. Related to the next comment, we also added an introductory sentence before presenting the data over the France domain and the global atmospheric forcing.

P.14, L430
« It is known that biases can emerge in simulated surface/sub-surface variables in response to specific atmospheric conditions, therefore different forcing datasets were used in the study. More specifically, an extensively validated high-resolution atmospheric forcing over France was preferred to coarser global forcing that may influence hydrological responses in a negative way, especially considering the large topographic variability over France. This limits the comparison between watersheds situated in France and other basins, but it gives more credit to the results within between similar watersheds. »

L483: Biases may appear in simulated surface/sub-surface variables

This sentence is confusing, as this is "excuse" to use multiple forcing datasets. It is better to state that "multiple forcing datasets are used in this study" for better under-standing of this paragraph.

Related to the previous comment, this sentence has been moved to the introductory/first sentence of this paragraph.

L513: with an average discharge increased by 0.7 %

Is it reasonable that the average discharge increased? What was the background physics mechanism? Please explain. I guess this could be due to the change in discharge timing, and it changes the discharge at the start and/or end of the simulation period resulting in a slight difference in the total discharge. In this case, the 0.7%increase is negligible and better to state "average discharge is not affected" rather than"0.7% increased".

As mentioned in your comment, the increase of the average discharge is linked to a temporal shift induced by the lag-effect related to the lake water dynamics over the simulation period (which is the same for all simulations). The sentence has been changed with:

P.15, L485
"Lake Geneva reduces the Rhône river discharge variability on average by 22 %."

L525: where the weir width is increased by a factor of five compared to ctrip_mlake_w1.

I wonder whether this simulation setting (500% of the control experiment) is reasonable or not. Given that the "lake outlet width" is observable parameter, there must be a reasonable range for this parameter. I think the sensitivity test should be designed within this "reasonable parameter range".

Thank you for raising this interesting question.

At the time of the study, we did not have access to any information on the lake outlet width. In this context, we decided to set a range for this parameter. Even if increasing by a factor five seems to be out of range for a lake outlet, this choice is motivated by the need to look at extreme values in order to find limits in the behaviour of our model. Using such an extreme range is valuable to show that in some situations, even a larger width does not help in finding an appropriate hydrograph (for example, at the outlet of Lake Victoria).

L620: lake level variations (sigma_s = 1069 m3s−1, sigma_o = 1003 m3s−1).

Is the unit (m3/s) correct? As this sentence mentioned for lake level.

Thank you for pointing out this typo. The unit is correct, however the related text is not. Lake level must be changed with "observed river discharge".

L633: The NIC score has…

What is "NCC score"? It once appears in the abstract first, and appears first time herein the maintext without any definition or explanation. Please provide the description of NIC score.

The description of NIC score is detailed in the Appendix. The reference has been added P19. L599.
L666: the worlds largest freshwater continental water body

Please explain "largest in terms of what?"

Sentence has been changed to:
"The Angara basin is dominated by Lake Baikal which is the world's largest freshwater continental reservoir"

L693: with a net decrease in the peak discharge.

What does "net decrease in the peak discharge"? Usually the term "net" is used for the total summation, but the term is used for "peak (maximum)" here. How can we define"net decrease in the peak"?

We agree on your comment and the inadequacy of writing *net* while talking about a peak discharge. The modified sentence uses the word "significant" instead of "net".
"with a significant decrease in the peak discharge"

L717: Simulations reveal the capability of the non-calibrated CTRIP-Mlake

This paragraph mainly discusses the impact of lake internal dynamics caused by wind, rather than discussing the overall limitation of the proposed model. The lake internal dynamic parts is better to be shown as a separate sub-section (i.e. 6.1 lake internal dynamics), as the following discussion points are explicitly shows with sub section title. I also suggest that the most significant difficulty of the internal height variation appears in the comparison between modeled and observed water levels. This point should be discussed explicitly. In addition, freezing of lake surface could cause significant difference in simulations. Is this represented in the current model? If not, better to explain as one of the major limitations.

We agree on the need to modify the organisation of this section. As proposed, we separated the sub-section with the title "lake internal dynamics". Regarding the discussion point on the comparison between modeled and observed water levels and adding the fact the Referee#2 also pointed out the need to provide more of an explanation in this section, the following paragraph has been added:

P.22, L695:
" One of the easiest approaches could  be to also take into account simple bathymetry in order  to  characterise  a  distributed  water  layer.  Modelling  could  also  benefit  from observations  datasets.  As  was  done  for  lake  Geneva,  these  gaps  could  be  overcome  by gathering data from several measurement sites  along the lake shore, but this depends on the  data  availability.  Over   the  long-term,  comparison  between  modeled  and  observed water levels could be improved by valuable satellite data as proposed in the Surface Water and Ocean Topography (SWOT, Biancamaria et al, 2016)."

Concerning the interesting point of the freezing of the lake surface:  currently MLake is not considering  this process. That's why the following paragraph has been added to the last sub-section of the discussion "Coupling MLake to the SURFEX modelling platform"

P.25, L811:
Furthermore, the coupling will also improve the simulation of lake surface freezing which remains one of the major limitations that could influence MLake. In the current version, only Flake explicitly represents frozen lakes in terms of the lake surface energy budget.

L764: Historical Lake Victoria level drops

I think it is better to remove this "sub-section title", to keep the consistency with other paragraphs.

We agree on the need to respect the consistency which is why we added the proposed unnumbered paragraph title "General impacts" consisting in a discussion for lake Geneva and lake Baïkal and the unnumbered paragraph title "Closer look on the Lake Victoria historical level drops" for lake Victoria.

L865: the lake outlet size

The term "lake outlet size" only appears here. Please use "lake outlet width" to keep consistency.

The sentence has been corrected.

L871: Finally.
It is a bit strange to see "Finally" after the sentence starting from "Last but not least".

The sentence has been corrected.

Figure 3: Please change the figure caption from French to English.

The caption has been changed and the figure is the figure 4.

Figure 4:It is difficult to know what the authors want to discuss with this figure. Please explain what does this figure want to explain in the caption, for better understanding of the river/lake map preparation.

We propose to change the figure into a more understandable figure that better explains the creation of the different lake masks and their integration on the river network. The new figure is now the number 4.

Figure 9: Is it possible to add observed discharge in this figure?

Figure 9 is intended to only show the influence of the lake mass balance effect on the CTRIP simulations. It is a prior evaluation. The observations are added in the Figure 10 for a comparison between modelled and observed river discharges.

---

## Author Comment (AC2) · 28 Jan 2021

The authors would like to thank the referee for the time allocated to the detailed revision of this paper and her/his positive comments about our work that help us to come up what we believe is an improved version of the manuscript. Our point by point responses to the reviewer comments are below in blue.

Please also note the supplement to this comment:
https://gmd.copernicus.org/preprints/gmd-2020-296/gmd-2020-296-AC2-supplement.pdf
* * *
[Figure]

2020.

**Supplement:**

Review for Parameterization of lakes water dynamics in the ISBA-CTRIP land surface system by Guinaldo et al.

Paper summary:
The paper describes new lake mass balance model component called Mlake and how the new module is integrated into the river routing model (CTRIP) coupled to ISBA land surface model. The paper also evaluated the developed model at 4 river basins across the globes and shows improvement of downstream discharges and lake level simulations compared to the observations.

Overall comments:

The descriptions of lake water balance model implementation into a river model for ESM fits very well in the GMD, model description paper. The lake water balance computation used in this paper seems to be more precise(account for lake-ground water interaction, separate treatment for surface runoff inflow and baseflow inflow) than other lake models even used for global hydrologic models. So, particularly for the ESM, I think it is substantial advancement of model process representations. Simulated impacts of lake parameter (i.e., outlet structure parameter) on the lake level and downstream discharge seems to be reasonable and discussed well. Figures showing results are also clear overall. My main concern is a lack of conciseness: the paper can be shortened by textural editing as well as by cutting some materials. More importantly, I feel the authors need to polish the descriptions throughout the paper. I put a few science related comments following numerous minor editorial comments I found as I read.

The authors would like to thank the referee for the time allocated to the detailed revision of this paper and her/his positive comments about our work that help us to come up what we believe is an improved version of the manuscript. Our point by point responses to the reviewer comments are below in blue.

Specific comments:

•P29-L920: This might be typo, but KGE expression seems to be incorrect. Standard KGE (Gupta et al., 2009) is a distance based on correlation(r), ratio of standard deviation (alpha) and mean ratio (beta). Modified KGE (from Kling et al. 2012) uses ratio of coefficient of variation (gamma) instead of ratio of standard deviation(alpha).In my opinion, KGE is convenient for model calibration since KGE aggregates three metrics into one, which allow modelers to use a single objective (target metric) to optimize the model, rather than multi-objectives, but for just model evaluation, KGE (aggregated metric) itself does not mean much(what aspects of time series the model simulate better or worse). Since table 6 compare Qs/Qo and sigma_s/sigma_o, I would suggest adding correlation and removing KGE.

Thank you for raising this point. We agree with you about the KGE expression and we have changed it accordingly. The modified KGE expression was used in the evaluation thus, the description is now corrected.
We also understand your concern about using KGE. This score have been replaced by the correlation in Table 6.

•Section 2.3.2 and 2.3.3 seems be a crux of the paper (descriptions of lake implementation strategy), and therefore need to be clear so reader can understand how exactly the lake are implemented in the network. Unfortunately, I am having hard time following section

2.3.2 after I read through several times. So,I feel I need to request the author revise this section drastically. Section 2.3.3 is described well.

The section has been modified as well as Fig. 3 and Fig. 4. The latest versions of the figures are now more readable and show the step described in the paragraph 2.3.2

•P22, L717: I found this (sub-grid variability of lake levels for a large lake) is interesting discussion and may be important. Impact of such lake sub-grid variability on downstream discharge is described in L725, but not clear and would be nice to see a little more elaboration on this. For even larger lakes (e.g., Great Lakes), the lakes should be represented by a number of grid boxes, and make individual grided lake interact each other?

We agree on the need to modify the clarify of this section. As proposed by Referee#1, we have separated a sub-section with the title "lake internal dynamics". Regarding the discussion point on the sub-grid variability, the following paragraph has been added:

P.22, L695:
" One of the easiest approaches could be to also take into account simple bathymetry in order to characterise a distributed water layer. Modelling could also benefit from observations datasets. As was done for lake Geneva, these gaps could be overcome by gathering data from several measurement sites along the lake shore, but this depends on the data availability. Over the long-term, comparison between modeled and observed water levels could be improved by valuable satellite data as proposed in the Surface Water and Ocean Topography (SWOT, Biancamaria et al, 2016)."

Editorial comments:

•Throughout the paper: The authors use "resolution",instead of "solution", which I would suggest using (e.g., numerical solution, not numerical resolution). Using "resolution" to mean "solve an equation or something" be confusing since "resolution" is also used for grid box sizes and vertical resolution of soil layer, snow layers

Thank you for raising this point. We agree that a mismatch has been made between the two definitions. We corrected the manuscript to include the appropriate wording (see L.150, L.205, L232, L236, L385, L388).

.•Throughout the paper: I know this may not be critical issue...but I feel the paper uses "scale" carelessly. "scale" is the characteristic length (space and time) of processes, modeling and observations (see Blöschl et al., 1995: https://onlinelibrary.wiley.com/doi/abs/10.1002/hyp.3360090305). For example, "global scale model" used throughout the paper means globe is the characteristic length of modeling, but it really means global domain model and scale used for model should be smaller.

The comment is justified since scale usually should correspond to some specific time (diurnal, decad, month…) or distance (mesoscale, beta, micro, kilometric etc…).
the authors went throughthe manuscript and made sure each time that the word "scale" was mentioned, some distance or time dimension/unit was attached to it.

Also we added this text (below) at the beginning of the manuscript in order to better describe what are we referring to when referring to scale:

L.185.
"In this study, we refer to CTRIP as a global scale model meaning that it is a 1/12° degree resolution model applied to areas ranging from large basins to a domain  covering the entire globe."

•Title: SURFEX v8.1 is the name of land surface model platform, but here, should lake model component name and version be in title. Here, is MLake name?

The names of the model version and the land surface model platform were added in order to respect the GMD specifications since Mlake is intended to be included in the latest SURFEX version. At the moment, there is no mention of a version for the lake model. However, the name of the lake model should be added to the title, thus we propose:

**Proposed title: "Parametrization of a lakes water dynamics model MLake in the ISBA-CTRIP land surface system (SURFEX v8.1)"**

•P2, L44-46: I feel this sentence is really opening sentence. Descriptions priors to this sentence seem to be little related to the topic in this paper.

Referee#1 raised the same issue. Thus, we deleted the first paragraph and changed the opening sentence with:

P.2, L28
« Only 2.5 % of the total water mass of the planet is defined as fresh water, and only a very small fraction is directly accessible for human consumption (Oki and Kanae, 2006). Lakes are of fundamental importance to ensure freshwater supply to the 800 million people which have insufficient safe drinking water, according to the World Health Organization (WHO, 2010; Marsily et al., 2018).»

•P3, L65: Suggest adding references .Climate change impact on Lake Chad has beenreportedin publication (https://www.nature.com/articles/s41598-020-62417-w)

Reference added: "Pham-Duc, B., Sylvestre, F., Papa, F., Frappart, F., Bouchez, C., and Crétaux, J.-F.: The Lake Chad hydrology under current climate change, Scientific reports, 10, 1–10, 2020."

•P3, L70: Suggest changing runoff -> discharge. e.g., lowering inter-annual and seasonal variability of downstream discharge?

The sentence has been changed.

•P3, L95: "General Hydrological Model" -> Global Hydrological Model?? If not, what does General Hydrologic model mean?

It is a mismatch between General Circulation Model and Global Hydrological Model. We changed the sentence to say  'Global Hydrological Model'.

•P5, L136: Awkward description. Suggest removing this.

We agree this paragraph is not essential and therefore we have removed it.

•P7, L214: main driver -> main motivation?

This has been corrected

•P8, L258: what does "component" mean here? I believe it is lake, but please be specific.

This has been corrected
P8. L244
"However, integrating a lake which can cover more than one grid cell in the CTRIP river networks is not straightforward"

•P8, L272: "dynamic close to a lake". Not clear to me.

This has been corrected as:
P8. L259:
"In some regions, the river stretch can be large and thus the streamflow time response remains slow which can be close to the response time of a lake"

•P8, L273: "lake hydrological dynamic". Sounds awkward to me.
This has been corrected as:
P9.260
"Consequently, finding a compromise between the lake spatial extension at different resolutions and
the actual lake water dynamic is important"

•section2.2: this section provides brief description of Flake model, which simulates energy balance in lake (my understanding), and lake evaporation is a part of the energy balance. I feel header should be "Flake model: lake energy balance model". This way, consistent with the following section (Mlake)

The sub-section title has been added.

•Section2.3.4: I am not sure about importance of this section. Figure 7 could follow more conventionofflowchart(https://en.wikipedia.org/wiki/Flowchart). I wonder if this section (after shortened) could be moved to very beginning of section 2.

In accordance with your comment, this section has been shortened and moved to the appendix section as it is useful to understand how the program works which we understand is within in the scope of GMD publications.

•P22, L719: "uni-dimensional" -> one-dimensional.

This has been corrected

•P22, L724: "fast time variations of the river discharge". Not clear phrase. Please consider describing different way.
Proposed sentence:

P.22, L.690

"Observed height differences over lakes can reach several meters from one shore to another depending on the wind stress and the distance of the fetch among other factors, and consequently this can influence the relatively high frequency variability of river discharge"

•P25, L798: "a unique composite energy budget for soil and vegetation". Does this mean control volume for energy budget computation is combined of soil and vegetation (not separate)?

The idea is correct. The default version ISBA/SURFEX used for this study uses a composite soil-vegetation layer for the non-snow covered surface energy budget, while an explicit snow scheme is used for the snow surface energy budget. This version has been the standard version in recent years for both hydrological studies and for our fully coupled CMIP6 global climate simulations. A more recent version has been developed which separates the soil from the vegetation but it was not finalized (fully evaluated) at the time this work was performed. But as mentioned in the conclusions, this version will be used in the future with the lake module and is expected to have an impact essentially in high latitude forest-dominated basins. But this will not impact the formulation of the lake model presented herein nor the conclusions of this work.

•P26, L834-836: this paragraph sounds out of place in this section (simulation sensitivity to lake outlet width)

A sub-section has been added in order to keep this discussion part consistent with the rest of the section.

•P27, L842: "has been conducted" -> use past tense?
This has been corrected

•P27, L842: "four river networks" -> four river basins?
This has been corrected

•P27, L865: "monitoring" -> monitor.
This has been corrected

•P27, L867-869: I would suggest moving this sentence to the end of the paragraph (replace the last sentence with this). Eventually we would like to see the effect of lake bathymetries on lake levels, downstream discharge across the globes.

Last sentence of the paragraph has been replaced by the following sentence:
P27.L852
"All this advocates for results to be extended to the global scale in order to characterise the systemic improvement for an ensemble of climate and physiographic conditions."

•P28, L878-L879: I would suggest adding references. Some groups have done some work on reservoir operation schemes, e.g., Hanasaki et al., 2006 JH, Shin et al., 2020 WRR.

The references have been added and the sentence has been modified as:

"Numerous studies have been focused on such developments (Hanasaki et al., 2006; Zhou et al., 2016; Busker et al., 2019; Shin et al., 2019) and on-going research is focusing

on creating a global reservoir system that will be added to MLake to improve the representation of dam operating rules."

•P28: Appendix header is missing?
This has been corrected.

•Figure 11. Add y-axis labels

The y-axis labels have been added to the figures.